# Simulating damage for wind storms in the land surface model ORCHIDEE-CAN (revision 4262)

Yi-Ying Chen[1,a,*], Barry Gardiner[2], Ferenc Pasztor[1,b], Kristina Blennow[3], James Ryder[1], Aude Valade[4], Kim Naudts[1,c], Juliane Otto[1,d], Matthew J. McGrath[1], Carole Planque[5], and Sebastiaan Luyssaert[1,e,*]

[1]Laboratoire des Sciences du Climat et de l'Environnement (LSCE/IPSL), CEA-CNRS-UVSQ, Université Paris-Saclay, Gif-sur-Yvette, France
[2]Institute National de la Recherche Agronomique (INRA), Villenave d'Ornon, France
[3]Swedish University of Agricultural Sciences (SLU), Alnarp, Sweden
[4]Institut Pierre Simon Laplace (IPSL), CNRS-UPMC, Paris, France
[5]CNRM/GMME/VEGEO Météo France,Toulouse, France
[a]now at: Research Center for Environmental Changes (RCEC), Academia Sinica, Taipei, Taiwan
[b]now at: Maritime Strategies International Ltd (MSI), London, England
[c]now at: Max Planck Institute for Meteorology, Hamburg, Germany
[d]now at: Climate Service Center Germany (GERICS), Helmholtz-Zentrum Geesthacht, Hamburg, Germany
[e]now at: Department of Ecological Sciences, Vrij Universiteit Amsterdam, Amsterdam, the Netherlands
[*]Equal contributions

*Correspondence to:* Yi-Ying Chen (yiyingchen@gate.sinica.edu.tw)

**Abstract.** Earth System Models (ESMs) are currently the most advanced tools with which to study the interactions between humans, ecosystem productivity and the climate. The inclusion of storm damage in ESMs has long been hampered by their big-leaf approach which ignores the canopy structure information that is required for process-based wind throw modelling. Recently the big-leaf assumptions in the large scale land surface model ORCHIDEE-CAN were replaced by a three dimensional description of the canopy structure. This opened the way to the integration of the processes from the small-scale wind damage risk model ForestGALES into ORCHIDEE-CAN. The integration of ForestGALES into ORCHIDEE-CAN required, however, developing numerically efficient solutions to deal with: (1) landscape heterogeneity, i.e., account for newly established forest edges for the parametrisation of gusts; (2) downscaling spatially and temporally aggregated wind fields to obtain more realistic wind speeds that would represents gusts; and (3) downscaling storm damage within the $2500\,km^2$ pixels of ORCHIDEE-CAN. This new version of ORCHIDEE-CAN was parametrised over Sweden. Subsequently, the performance of the model was tested against data for historical storms in Southern Sweden between 1951 and 2010, and South-western France in 2009. In years without big storms, here defined as a storm damaging less than $15 \times 10^6\,m^3$ of wood in Sweden, the model error is $1.62 \times 10^6\,m^3$ which is about $100\,\%$ of the observed damage. For years with big storms, such as Gudrun in 2005, the model error increased to $5.05 \times 10^6\,m^3$ which is between $10\,\%$ and $50\,\%$ of the observed damage. When the same model parameters were used over France, the model reproduced a decrease in leaf area index and an increase in albedo, in accordance with SPOT-VGT and MODIS records following the passing of Cyclone Klaus in 2009. The current version of ORCHIDEE-CAN (revision 4262) is therefore expected to have the capability to capture the dynamics of forest structure due to storm disturbance

both at regional and global scales, although the empirical parameters calculating gustiness from the gridded wind fields and storm damage from critical wind speeds may benefit from regional fitting.

# 1 Introduction

During the last 15 years, Western Europe has been severely affected by storms with the six most damaging European storms
ever recorded hitting France (Lothar and Martin, December 1999; Klaus in January 2009), Sweden (Gudrun, January 2005)
and Germany (Lothar, December 1999; Kyrill, January 2007) as well as neighbouring countries (the Netherlands, Belgium,
Switzerland, Czech Republic, Slovakia and the Baltic States) (Gardiner et al., 2010). The short-term impact of wind on forests
includes billions of Euro of damage to wood stocks, loss of valuable protected old forest stands, increased fire occurrence
(Miller et al., 2007), pest vulnerability (Komonen et al., 2011), and a temporary decrease in productivity of the remaining forest
stands (Merrens and Peart, 1992; Everham et al., 1996; Seidl and Blennow, 2012). Wind throw is reported to be the cause of
57 % of forest disturbances in Europe and is thus more significant than stand replacing disturbances through pest attack (26 %)
and through fire (16 %) (Schelhaas et al., 2003; Seidl et al., 2014). Furthermore, a global literature review indicates that wind
disturbance triggers fire activity in warm and dry climates but induces pathogens/insect outbreak in warm and wet climates
(Seidl et al., 2017).

Similar to fire (Randerson et al., 2006), the direct and indirect effects of wind disturbances may contribute to the top of
the atmosphere radiative forcing (O'Halloran et al., 2012). The direct effects, such as a reduction in leaf area index (Juárez
et al., 2008), transpiration (Negrón-Juárez et al., 2010), an increase in the surface albedo (Planque et al., 2017), and decrease
in roughness (Zhu, 2008) have been shown to impact the regional climate, i.e., following the storm Klaus in 2009, cloud
cover frequency was observed to decrease over Les Landes in Southwestern France (Teuling et al., 2017). The indirect effects
include a reduction of the gross productivity by damage to the rooting system, increased tree mortality due to facilitating
insect or pathogen outbreaks (Sturrock et al., 2011), or a change of the forest ecosystem by shifting canopy structure (Lin
et al., 2017). Storm-induced disturbances are likely to provide feedbacks on climate through direct effects such as increasing
greenhouse gas emissions (Lindroth et al., 2009), increasing surface albedo (Planque et al., 2017), and decreasing local cloud
frequency (Teuling et al., 2017) as well as indirect effects such as increased natural disturbances, a reduced logging rate in
subsequent years, increased weathering, and increased C, N and cation leaching (Futter et al., 2011; Köhler et al., 2011).
Increased weathering and leaching could even extend the effects of wind throw from the land to the oceans since terrestrial
processes have been found to play an important role in the lateral C fluxes to the oceans through inland waters (Battin et al.,
2009; Regnier et al., 2013).

ESMs can be seen as a mathematical representation of the major biophysical processes of the natural world (Sellers et al.,
1986; Henderson-Sellers et al., 1996; Sellers, 1997; Bonan, 2008) and are currently the most advanced tools to study the inter-
actions between humans, their use of vegetated ecosystems, and the climate (Jackson et al., 2005; Swann et al., 2012; Naudts
et al., 2016). Although Earth system modelling groups dedicate considerable resources to studying the effects of fires (Lasslop
et al., 2014; Yue et al., 2017), forest management (Naudts et al., 2016), land cover changes (Swann et al., 2012; Devaraju et al.,

2015), and shifting cultivation (Wilkenskjeld et al., 2014), storm induced disturbances and their climate feedback are not yet explicitly dealt with in ESMs. The objective of this study is to develop the model capability for the ESM IPSL-CM, through its land component ORCHIDEE-CAN, to simulate the effects of wind storms on the land surface by building on a good understanding of ecosystem level processes (Hale et al., 2012, 2015). Until the direct and indirect climate effects of wind storms have been quantified, implementing storm damage in ESMs is justified by its precursory effect on other natural disturbances such as fires and insects (Seidl et al., 2014). Abrupt mortality from drought, wind storms, fires, pests and their interactions will need to be accounted for, if ESMs are to be used to quantify the effects of future climate on forest dynamics and forest resilience.

Several classification mostly based on maximum wind speeds within a cyclone have been proposed to define different types of storms ranging from depressions to hurricanes. In general, storm damage strongly depends on the frequency and intensity of storms. For example, when gusts within a cyclone exceed $20 \ ms^{-1}$, uprooting and stem breakage is to be expected resulting in severe damage. When wind speeds remain below $17.1 \ ms^{-1}$ the damage is expected to be less devastating but nevertheless substantial (Scatena et al., 2004). The implemented approach, however, did not require a classification of storms as it can simulate the transition from no storm damage at low wind speeds to a stand replacing disturbance at high wind speeds (2.2.4).

## 2 Model description and parametrisation

ORCHIDEE-CAN revision 2566 (see section 2.1) was further developed by implementing the modifications and additions listed below (see sections 2.2 to 2.3), resulting in revision 4262. From revision 4262 onwards, ORCHIDEE-CAN has the capability to simulate tree mortality from wind storms. The notation used to describe the model is listed in full in Table 1.

### 2.1 ORCHIDEE-CAN (revision 2566)

ORCHIDEE (Krinner et al., 2005; Bellassen et al., 2010) is the land surface model of the IPSL (Institute Pierre Simon Laplace) Earth System Model. Hence, by conception, it can be coupled to a global circulation model. In a coupled set-up, the atmospheric conditions affect the land surface and the land surface, in turn, affects the atmospheric conditions. However, when a study focuses on changes in the land surface rather than on the interaction with climate, It also can be run off-line as a stand-alone land surface model. The stand-alone configuration receives the atmospheric conditions such as temperature, humidity and wind, to mention a few, from the so-called "forcing files". Unlike the coupled set-up, which needs to run at the global scale, the stand alone configuration can cover any area ranging from the global domain to a single grid point.

Although ORCHIDEE does not enforce a spatial or temporal resolution, the model does use a spatial grid and equidistant time steps. The spatial resolution is an implicit user setting that is determined by the coarsest resolution of the forcing data and the boundary conditions, i.e., the vegetation distribution, climatological forcing data, and the soil map. If higher resolution drivers are available the model can then be run at that scale. If site-level drivers are available then simulations at the site scale are feasible. ORCHIDEE can run on any temporal resolution, however, this apparent flexibility is rather restricted as the processes are formalized at given time steps: half-hourly (i.e. photosynthesis and energy budget), daily (i.e. net primary

production) and annual time step (i.e. vegetation dynamics). Hence, meaningful simulations have a temporal resolution of 15 minutes to one hour for the energy balance, water balance and photosynthesis calculations.

ORCHIDEE builds on the concept of meta-classes to describe vegetation distribution. By default it distinguishes 13 such meta-classes (one for bare soil, eight for forests, two for grasslands and two for crop lands). Each meta-class can be subdivided
in an unlimited number of Plant Functional Types (PFTs). When simulations make use of species–specific parameters and age classes (as is the case in this study), several PFTs belonging to a single meta-class will be defined. Biogeochemical and biophysical variables are calculated for each PFT.

ORCHIDEE-CAN (revision 2566) (Naudts et al., 2015; Ryder et al., 2016; Chen et al., 2016; McGrath et al., 2016) is one of the branches for the ORCHIDEE model development, which was selected to simulate large-scale wind throw and storm
damage because, contrary to most land surface models, ORCHIDEE-CAN simulates dynamic canopy structures, a feature essential to simulate the likelihood of wind throw and the subsequent damage. Changes in canopy structure resulting from wind throw are then accounted for in the calculations of the carbon, water and energy exchange between the land surface and the lower atmosphere.

In ORCHIDEE-CAN, tree height and crown diameter are linked to tree diameter through allometric relationships. Individual
tree canopies are simulated as spherical elements with their horizontal location following a Poisson distribution across the stand (Naudts et al., 2015). A forest is represented by a user-defined number of diameter classes. Each diameter class represents trees with a different mean diameter and height and, therefore, informs about the social position of trees within the canopy. The difference in social position within a stand is the basis of intra-stand competition which accounts for the fact that trees with a dominant position in the canopy are more likely to intercept light than suppressed trees, and, therefore, contribute more to the
stand level photosynthesis and biomass growth (Deleuze et al., 2004). The allocation scheme is based on the pipe model theory (Shinozaki et al., 1964) and its implementation by Sitch et al. (2003). The scheme allocates carbon to different biomass pools (leaves, fine roots, and sapwood) while respecting the differences in longevity and hydraulic conductivity between the pools (Naudts et al., 2015).

At the start of a simulation, each PFT contains a user-defined number of diameter classes. This number is held constant,
whereas the boundaries of the classes are adjusted throughout a simulation to accommodate temporal evolution in the stand structure. By using flexible class boundaries with a fixed number of diameter classes, different forest structure can be simulated. An even-aged forest, for example, is simulated with a small diameter range between the smallest and largest trees. All trees thus belong to the same stratum. An uneven-aged forests is simulated by applying a large range between the diameter classes. Different diameter classes will, therefore, represent different strata. Each diameter class contains a single modelled
tree. Modelled tree is replicated to give realistic stand densities. Following this, tree growth, canopy dimensions and stand density are updated. Throughout a simulation, individual tree mortality causes stand density to decrease. In ORCHIDEE-CAN individual tree-mortality is caused by self-thinning, and forest management. In the absence of these processes, a constant rate of so-called environmental background mortality is applied. The inclusion of the so-called environmental background mortality implicitly accounts for mortality through fires, pests and wind throw. Following the development of the wind throw and storm

damage module in revision 4262, mortality from wind throw is now explicitly accounted for and thus no longer included in the so-called environmental background mortality.

Furthermore, age classes are used after land cover change and forest management events to simulate explicitly the regrowth of the forest. Following a land cover change, biomass and soil carbon pools (but not soil water columns) are either merged or split to represent the various outcomes of a land cover change. This dynamic approach to stand and landscape structure is exploited in other parts of the model, i.e. precipitation interception, transpiration, energy budget calculations, the radiation scheme, absorbed light for photosynthesis, and, since revision 4262, tree-mortality through wind storms.

## 2.2 ORCHIDEE-CAN (revision 4262)

In ORCHIDEE-CAN (revision 4262) the biomass of the different pools, leaf area index, crown volume, crown density, stem diameter, stem height and stand density are simulated as the accumulated growth, and are passed to the windthrow module. The windthrow module calculates the critical wind speed based on the principles applied in ForestGALES (Gardiner et al., 2000), and storm damage based on the approach developed and tested by Anyomi et al. (2017). Figure 1 summarises the major components of storm damage calculations in ORCHIDEE-CAN. A more detailed description of the different components in the figure is presented in following sections.

### 2.2.1 Critical wind speeds (ForestGALES)

The presence or absence of storm damage in a forest stand can be modelled with the concept of critical wind speed. If the wind speed exceeds the critical wind speed of a forest, the forces applied by the wind speed are sufficient to overturn the whole tree or break its stem. The exact value of the critical wind speed depends on the canopy structure (Vollsinger et al., 2005; Hale et al., 2012), the tree species, the soil properties and the root profiles (Nicoll et al., 2006). In this study, the physics formalized in ForestGALES (Gardiner et al., 2010; Hale et al., 2015), a hybrid mechanistic forest wind damage risk model, were added into ORCHIDEE-CAN (revision 4262) to simulate the critical wind speeds of all forest stands. The model simulates the critical wind speed for two types of damage: tree overturning and stem breakage. The critical wind speed for overturning is calculated as

$$CWS_{ov} = \frac{1}{\kappa D} \left( \frac{C_{reg} \cdot SW}{\rho G d} \right)^{\frac{1}{2}} \left( \frac{1}{f_{CW} \cdot f_{edge}} \right)^{\frac{1}{2}} ln \left( \frac{h-d}{z_0} \right) \tag{1}$$

Where $CWS$ is the critical wind speed (m s $^{-1}$), and the subscript $ov$ denotes the critical wind speed for overturning. $\kappa$ is von Karman constant and $D$ is the inter-tree spacing (m). $C_{reg}$ is a regression coefficient that was derived from tree pulling experiments (N $\cdot$ m kg$^{-1}$; Nicoll et al. (2006)). $SW$ is the green weight of the bole of the tree (kg). $SW$ is calculated by multiplying the model simulated above-ground biomass with green density for different tree species (see Table 2). $f_{CW}$ is the enhanced momentum caused by the overhanging displaced mass of the canopy. In ORCHIDEE-CAN $f_{CW}$ was set to 1.136 (unitless), as suggested by Nicoll et al. (2006) from analysing extensive tree-pulling data. In other words, the applied turning moment is a constant fraction of the total turning moment. $f_{edge}$ is the edge factor (unitless) and a detailed description of the factor is given in the section 2.2.2. $G$ is the gust factor (unitless) and its calculation is also described in section the 2.2.2. $h$

is the average tree height (m), $d$ is the zero-plane displacement (m), $z_0$ is the roughness length (m) and all are simulated by
ORCHIDEE-CAN (Naudts et al., 2015). Note that $d$ and $z_0$ depend on the wind speed at canopy height $u_h$, hence iterations
are required to solve Eq.(1) (see below).

The critical wind speed for stem breakage is calculated as

$$CWS_{bk} = \frac{1}{\kappa D} \left( \frac{\pi \cdot MOR \cdot dbh^3}{32\rho G(d-1.3)} \right)^{\frac{1}{2}} \left( \frac{f_{knot}}{f_{CW} \cdot f_{edge}} \right)^{\frac{1}{2}} ln\left( \frac{h-d}{z_0} \right) \tag{2}$$

Where the subscript $bk$ denotes stem breakage, $MOR$ is the modulus of rupture (Pa) of green wood and was parametrised for
different tree species (see Table 2) . $dbh$ (m) is the tree diameter at breast height as simulated by ORCHIDEE-CAN and $f_{knot}$
is a factor to reduce wood strength due to the presence of knots (unitless). Similar to Eq.(1), $d$ and $z_0$ in Eq.(2) also depended
on the wind speed measured at canopy height $u_h$, hence iterations are required to solve Eq.(2) (see below)

The relationships between the aerodynamic parameters ($d$ and $z_0$) and the vegetation structure follow the analytical relation-
ships proposed by Raupach (1994):

$$d(u_h) = h(1 - (\frac{1 - e^{-\left(c_{d1}\left(\frac{C_w \cdot C_d}{D^2}\right)C \cdot u_h^{-n}\right)^{\frac{1}{2}}}}{\left(c_{d1}\left(\frac{C_w \cdot C_d}{D^2}\right)C \cdot u_h^{-n}\right)^{\frac{1}{2}}})) \tag{3}$$

$$z_0(u_h) = (h-d)(e^{-\kappa \cdot \gamma + \Psi_h}) \tag{4}$$

Where $c_{d1}$ is a regression parameter ($c_{d1} = 7.5$), $C_w$ is crown width (m) and $C_d$ is crown depth (m). Individual trees in the
model are simulated as spherical elements, the canopy width and canopy depth are thus identical. $\gamma$ is a parameter that depends
on the canopy characteristics ($\gamma = \frac{1}{(0.003 + 0.15 \cdot \frac{C_w \cdot C_d}{D^2})^{1/2}}, max(\frac{C_w \cdot C_d}{D^2}) = 0.6$), $\Psi_h$ is the atmospheric stability correction func-
tion ($\Psi_h = ln(2) - 1 + \frac{1}{2}$).

Tree crowns, branches and stems are considered as porous and flexible materials that will streamline and thus change their
shape with changing wind speed ($u_h$). Streamlining was parametrised through the parameters $C$ and $n$, which were reported for
wind tunnel experiments with different tree species (Rudnicki et al., 2004; Vollsinger et al., 2005)(see Table 2). The maximum
value of $C_D = C \cdot u_h^{-n}$ is set at $u_h = 10\,ms^{-1}$ and the minimum value of $C_D$ is set at $u_h = 25\,ms^{-1}$ (limits are based on wind
speed range in Mayhead (1973)). The species specific streamlining effect for wind speeds outside of this range was calculated
be holding $u_h$ constant to its lower or higher threshold. In other words, $C_D$ was implemented as a kind of step function.

Critical wind speeds are calculated as the solutions of a non-linear set of equations for overturning, i.e., Eqs.1, 3 and 4,
and another set of equations for stem breakage, i.e., Eqs.2, 3 and 4. An initial wind speed, $u_h = 25\,ms^{-1}$, is applied to Eq.(3)
and Eq.(4) to obtain an approximation for $CWS_{ov}$ by applying Eq.(1). Similarly, an initial wind speed of $u_h = 25\,ms^{-1}$, is
applied to Eq.(3) and Eq.(4) to obtain an approximation for $CWS_{bk}$ by Eq.(2). Subsequently, $u_h$ is set to the value of $CWS_{ov}$
(or $CWS_{bk}$) to estimate the aerodynamic parameters ($d$ and $z_0$) for the next iteration. The iteration process is stopped if the
difference in $CWS$ between two iteration falls below $0.01\,ms^{-1}$ or the number of iterations exceeds a threshold of 20.

Whereas ORCHIDEE-CAN is designed to simulate both even and uneven-aged stands, ForestGALES is currently limited to
simulate the critical wind speeds for even-aged forests. Although this difference in design is thought to have few consequences,

it is considered essential in the calculation of the ratio between tree height and tree spacing (the so-called inter-tree spacing, $D$). In an even-aged stand both tree spacing and tree height are homogeneous and thus well-defined at the stand level. This is no longer the case for tree height in uneven-aged stands. This issue is accounted for by calculating a critical wind speed for each diameter class separately. Although, this approach addresses the possible heterogeneity in tree height, it requires a value

for inter-tree spacing, by definition a stand characteristic, to be calculated for each diameter class. To calculate the inter-tree spacing for each diameter class, firstly the total woody biomass at the stand level is calculated. Subsequently, this total woody biomass is divided by the biomass of the modelled trees in each diameter class. The outcome is considered the virtual inter-tree spacing of the diameter class $D$ and was used in Eq.(1) and Eq.(2) to calculate the critical wind speeds for each diameter class.

By default three diameter classes are used to describe the heterogeneity within a forest stand. ORCHIDEE-CAN then calcu-

lates the critical wind speed for breakage and overturning based on the vegetation structure parameters, for each diameter class. When using three diameter classes, as is the case in this study, a total of six critical wind speeds are thus calculated for each forest in each grid box. Subsequently, the lowest critical wind speed is used to determine the damage type for each diameter class. The number of damaged trees in each diameter class is then calculated by multiplying the damage rate ($D_\beta$; see section 2.2.4) with the tree numbers within each diameter class. The total number of trees damaged by a storm was calculated as the

sum of the damaged trees in each diameter class.

### 2.2.2  Gustiness and edge effect (ForestGALES)

ORCHIDEE-CAN is driven by half-hourly wind fields. For storm damage, such a time step is already too large because the half-hourly wind field hides the extreme wind gusts that occur within a half-hourly time step. Storm damage is determined more by the extreme gusts than by the average wind speed. This scaling issue is dealt with by explicitly simulating the gustiness

through the so-called gust factor. The gust factor $G$ was parametrised as a function of inter-tree spacing to tree height ratio, and edge distance to the tree height ratio. These dependencies and their parameter values are based on wind tunnel experiments (Gardiner et al., 2000; Hale et al., 2012, 2015).

$$G = \left( (-2.1 \cdot \frac{D}{h} + 0.91) \cdot \frac{x}{h} + (1.0611 \cdot ln(\frac{D}{h}) + 4.2) \right) \cdot G_{adj} \tag{5}$$

where $\frac{D}{h}$ is constrained between 0.075 and 0.45. $x$ is parametrised as a function of leaf area index (LAI), i.e., $x = \frac{28h}{LAI}$ for the

inner area nearby recent gaps and $x = 9h$ for the outer area forest away from such gaps (see section 2.2.3). The length scale parameter $x$ was derived from large eddy simulation (Pan et al., 2014). $G_{adj}$ is a linear scaling factor introduced to make the predicted gusts (based on wind tunnel experiments) better match the observed field measurements.

The edge effect was considered at the landscape level. Forests within a modelled grid were separated into two regions, inner area and the outer area, which can be simulated by the model (see Figure S1 in the supplementary). At the inner area, the effect

of vegetation structure on wind speed is accounted for through the edge factor $f_{edge}$ in Eq.(1) and Eq.(2). The calculation of $f_{edge}$ follows the approach proposed by Gardiner et al. (2000).

$$f_{edge} = \frac{(2.7193(\frac{D}{h}) - 0.061) + (-1.273(\frac{D}{h}) + 0.9701) \cdot (1.1127(\frac{D}{h}) + 0.0311^{\frac{x}{h}})}{(0.68(\frac{D}{h}) - 0.0385) + (-0.68(\frac{D}{h}) + 0.4785) \cdot (1.7239(\frac{D}{h}) + 0.0316^{\frac{x}{h}})} \tag{6}$$

At the outer area, the edge effect is negligible such that $f_{edge}$ is set to 1.0.

### 2.2.3 Vegetation structure

Vegetation structure is simulated at the landscape and the stand level (see Fig. 1). At the landscape level the simulations distinguish between forests with a newly formed forest edge and forest with established edges. Edges result from natural or anthropogenic stand replacing disturbances. First, the surface area of stand replacing disturbances is cumulated over the last five years ($A_5$), a time horizon corresponding to the time required for forests nearby newly formed edge to adapt to the increased gustiness (Gardiner and Stacey, 1996). By prescribing the average gap size ($A_{gap}$) to 2 ha, and assuming gaps are square shaped and the gustiness is affected over a distance of 9 times the canopy height ($9h$) (Gardiner and Stacey, 1996), the forest area that experiences an increased gustiness due to proximity of recent gaps ($A_{inner}$) is calculated as :

$$A_{inner} = \frac{1}{4} \cdot \left( \left( A_{gap}^{\frac{1}{2}} + 2 \cdot 9h \right)^2 - A_{gap} \right) \cdot \left( \frac{A_5}{A_{gap}} \right) \tag{7}$$

Where the factor of $\frac{1}{4}$ accounts for the fact that only the downwind edge perpendicular to the wind will experience an increased gustiness. The second term is the inner area generated by a single gap and the third term is the total number of gaps in the grid cell. The forest area that has no edges in its proximity $A_{outer}$ is calculated as the residual:

$$A_{outer} = \begin{cases} A_{grid} - (A_5 + A_{inner}), \text{ when } A_5 + A_{inner} < A_{grid} \\ 0 \text{ and } A_{inner} = A_{grid}, \text{ when } A_5 + A_{inner} \geq A_{grid} \end{cases} \tag{8}$$

Where $A_{grid}$ is the area of the grid box being modelled.

### 2.2.4 Storm damage

With wind speeds approaching the critical wind speed, damage such as defoliation and branch damage are becoming more likely. Once the wind speed exceeds the critical wind speed, overturning and stem breakage are possible but their likelihood increases with further increasing wind speeds. A sigmoid damage function is used to simulate the rate of storm damage to a forest. A similar approach has been applied and tested by Anyomi et al. (2017) for estimating storm damage as a function of the daily maximum wind speed. This relationship is formalized as :

$$D_\beta = D_{max} \left( \frac{1}{1 + e^{-\left( \frac{U_{max} - CWS_{bk,ov}}{R_f} \right)}} - \frac{1}{1 + e^{\left( \frac{CWS_{bk,ov}}{R_f} \right)}} \right) \tag{9}$$

Where $D_\beta$ is the damage rate (unitless) and thus the share of trees that will be killed, $D_{max}$ is an observed maximum damage rate which was set to 1.0. $R_f$ is a relaxation parameter to adjust the damage rate given by a certain wind speed below the model calculated critical wind speed, and a value 6.0 was applied for all tree species. $U_{max}$ is the maximum daily wind speed from the forcing or model calculation. Subsequently, the lowest out of the six calculated critical wind speeds (see section 2.2.1), is used to determine the damage type for each diameter class. The number of damaged trees in each diameter class is then calculated by multiplying the damage rate ($D_\beta$) with the tree numbers within each diameter class. The total number of trees damaged by a storm was calculated as the sum of the damaged trees in each diameter class.

### 2.2.5 Salvage logging

Damaged woods due to storms are often left on site in unmanaged forests, however the salvage logging is often carried out for a managed forest in order to recover some of the economic losses and avoid large scale insects outbreaks triggered by wind disturbance. When dealing with the effects of wind damage on the biomass pools of forests, the anthropogenic response to storm
damage needs to be accounted for. ORCHIDEE-CAN distinguishes managed and unmanaged forest. In unmanaged forests all carbon contained in trees killed by wind storms end up in the litter pools. For managed forests, a prescribed harvest ratio determines the wood that is salvaged logged and the wood left on site. In species that are prone to bark beetle attacks following wind throw, the volume left on site is very small. In Sweden, a maximum volume of $5\,m^3ha^{-1}$ newly damaged logs allowed. However, following large-scale storm damage this threshold has been temporarily lowered to $3\,m^3ha^{-1}$ in order to reduce the
risk of spruce bark beetle outbreaks. Following Gudrun, it was observed that no more than $1.8\,m^3ha^{-1}$ to $1.1\,m^3ha^{-1}$ was left on site. Given that the current implementation of storm damage was designed to deal with large wind storms, with a fair risk for subsequent bark beetle outbreaks (Kärvemo, 2015), applying a very high efficiency for salvage logging, i.e., 99%, appears justified Schroeder et al. (2006)).

### 2.2.6 Soil characteristics

Although ORCHIDEE-CAN distinguishes three soil texture classes, i.e., sand, clay, and loam, the current approach to simulating soil water hydrology, de Rosnay (2003) considers all soils to be free draining at the bottom of the soil layers. This assumption differs from ForestGALES where four soil classes with different drainage are distinguished (Hale et al., 2015): freely draining mineral soil, gleyed (i.e. waterlogged and lacking in oxygen) mineral soil, peaty mineral soil, and deep peat. At present, ORCHIDEE-CAN only uses the ForestGALES parameters for freely draining mineral soils. Owing to this assumption
ORCHIDEE-CAN is expected to overestimate the critical wind speed, resulting in less damage, for locations with shallow and/or wet soils.

Furthermore, ForestGALES distinguishes shallow, medium and deep rooting species. This classification was applied in ORCHIDEE-CAN through the parameter ($humcte$) describing the vertical root-profile. In ORCHIDEE-CAN, the rooting density is assumed to following a function that decreases exponentially from the top to the bottom of the soil layers and is
considered independent from site conditions or stand age. If 90 % of total root mass was found above a depth of 2 meters, the species was considered shallow rooted. The effect of rooting depth on critical wind speeds was accounted for by using a different regressing coefficient $C_{reg}$ for shallow and deep rooting species (see Table 2). Under the current parameter settings of the rooting profile, ORCHIDEE-CAN considers all tree species to be shallow rooted. Note that in ORCHIDEE-CAN the rooting profile is also a critical parameter with which to calculate drought stress of trees.
When the soil is frozen, ORCHIDEE-CAN only allows wind storm damage from stem breakage. In other words, there is no overturning of trees when the soil is frozen. The soil temperature at $0.8\,m$ below the surface was used as the threshold to decide whether the soil was frozen or not.

### 2.3 Model parametrization

#### 2.3.1 Downscaling wind fields

In this study, simulations are forced by six hour CRU-NCEP climate reanalysis (Viovy, 2011; Maignan et al., 2011). The internal weather generator of the ORCHIDEE-CAN model interpolates this reanalysis to obtain the half-hourly (30 min) mean
wind speed used in the calculation of storm damage. Interpolation of the six hour reanalysis is expected to dampen the wind speed and, therefore, wind damage calculated by ORCHIDEE-CAN would be underestimated. To overcome this issue a tuning parameter called Mean Wind Ratio (MWR) was introduced.

The MWR converts the mean wind speed from six hour CRU-NCEP reanalysis into a mean wind speed at the 30 min time step. ORCHIDEE-CAN then uses these daily maximum estimated values to calculate damage rates that may occur in a forest
stand nearby or away from forest edges. Note that in this study the values for the mean wind ratio are specific for the CRU-NCEP half degree forcing at a six hour time step. The mean wind ratio will thus need to be re-parametrised if the wind driver is replaced by any other forcing.

The MWR was estimated from 38 European eddy-covariance sites covered by forests for which the meteorological data were freely available. For the period 1996 to 2007, 208 site-year combinations were retained for which over 60 % of the half
hourly measurements were available. The remaining data gaps were filled with the ERA-Interim reanalysis data (Dee et al., 2011; Vuichard and Papale, 2015). The 208 site-years were extracted from the CRU-NCEP reanalysis. Subsequently, the ratio between the maximum 30 min mean wind speed within a six hour period and the six hour mean wind speed obtained for the same location and time frame from the CRU-NCEP reanalysis was calculated.

$$MWR = \frac{U_{Fluxnet}}{U_{CRU-NCEP}} \tag{10}$$

Where the subscripts Fluxnet and CRU-NCEP denote the data source, and $U_{Fluxnet}$ is the maximum value of the 30 min mean wind speeds ($\mathrm{m\,s^{-1}}$) within a six hour time frame, and $U_{CRU-NCEP}$ is the six hour mean wind speed from the CRU-NCEP dataset. Finally, maximum wind speeds in the time series of each forest site were stratified according to the wind force catalogue in Beaufort Wind Scale (BWS), an empirical measure that relates wind speed to observed conditions on land (National Meteorological Library and Archive, 2010). Data were then analysed to account for the relationship between the
mean wind speed and the ratio between the observed and temporal average mean wind speed.

The quality of the fitted relationship was calculated as the root mean square error (RMSE). RMSE is a statistic that is widely applied to quantify the difference between values predicted by a model and the values actually observed:

$$RMSE = \sqrt{[mean(Y_i - F_i)]^2} \tag{11}$$

Where $Y$ denotes the observed values, $F$ denotes the modelled values, and the subscription $i$ is the sample index. RMSE is
used throughout this study to quantify the goodness of fit between observed and predicted values.

### 2.3.2 Critical wind speeds for five tree species

The calculation of critical wind speeds is parametrized for five common tree species in Europe: Norway spruce (*Picea sp.*), Scots pine (*Pinus sylvestris*), beech (*Fagus sylvatica*), birch (*Betula sp.*) and Oak (*Quercus ilex/subler*). In ORCHIDEE-CAN these five tree species are simulated as separate PFTs over Europe. Parameters for the other PFTs within and outside of

Europe were based on ForestGALES which has been parametrized for 21 tree species including the five species simulated by ORCHIDEE-CAN. Table 2 lists the parameters used in ORCHIDEE-CAN to calculate the critical wind speeds.

    Spruce, pine and birch make up almost the entire forest cover of Southern Sweden. Pine is the single most important species in Les Landes. These regions were only used to test the model. The anticipated simulation domain of this new development is Europe. The five species for which the model was tested make up 67 % of the European forest cover. In terms of taxonomic

families the representativeness increases to >90 % (Koeble and Seufert, 2001).

## 3 Observational data, model tuning, and evaluation

### 3.1 Storm damage observations

A 60-year long record of storm damage statistics over Sweden was extracted from the country-level European Forest Institute storm damage database from 1951 to 2010 (Nilsson et al., 2004; Schlyter et al., 2006; Bengtsson and Nilsson, 2007; Gardiner

et al., 2010). We refined this dataset with regional information. Following Gudrun in January 2005, the Swedish Forest Agency mapped the spatial distribution of the storm damage for damage classes ranging from 5 to 50 $m^3$ per ha in steps of 5 $m^3$ for the first damage class and 10 $m^3$ for all subsequent classes (See Fig.7). The storm made landfall near by the south of Norway and went through the north of Götaland and resulted in extensive forest damage in the central area of Götaland. The area around the cites of Ljungby and Växjö, was reported having the greatest damage of about 30 $m^3 ha^{-1}$. The spatial extent of storm damage

was retrieved from ocular inspection of aircraft images processed by the Swedish Forest Agency. In January 2009, Klaus made landfall in Southwesten France nearby Les Landes forest and damaged 43.1 $\times 10^6 \, m^3$ of wood in France. The dynamics of surface albedo and LAI between 2001 and 2010 were extracted from MODIS and SPOT-VGT satellite images, receptively (Planque et al., 2017). These remote sensing time series were overlaid by ORCHIDEE-CAN simulations and compared at two locations between 2001 and 2010 thus resulting in a total of 20 data points (shown as pink arrows in Figure S2).

### 3.2 Model tuning for storm damage

Although all parts of the wind throw module come with their own assumptions, parameters and subsequent uncertainties, the calculation of the gustiness ($G$; Eq.(5)) is considered to be among the most uncertain part of the model because it involves spatial and temporal scaling issues in both the driver data and the model formulation. Furthermore, the function that was used to convert the difference between the critical wind speed and wind speed into a damage rate ($D_\beta$; Eq.(9)) is also thought of as

very uncertain. The key parameters in these functions, $G_{adj}$ and $R_f$ are empirical, lack a good observational constraint and are therefore prime parameters to be tuned for matching the observations.

Given the availability of 60 years of observations of primary damage caused by wind storms between 1951-2010 for the region (Gardiner et al., 2010), southern Sweden was selected as the study area for model tuning. Tuning made use of the observed damage volumes for the years 1981 to 2000 because it is the most recent period without major storms. The storm Gudrun (2005) was deliberately excluded from the tuning period so it could be used as an evaluation of model performance.

The simulations used for parameter tuning started in 1981 and had, therefore, to be forced by a spatially explicit description of the biomass distribution in southern Sweden at the end of the year 1980. In this study, the spatially-explicit biomass distribution for 1980 was extracted from a previous study that simulated the effects of forest management and land cover changes in Europe between 1750 to 2010 (McGrath et al., 2015; Naudts et al., 2016). The previous study did, however, not account for the effects of wind storms on woody biomass and is, therefore, likely to overestimate the biomass in regions with chronic wind stress.

This issue was overcome by starting the simulation in 1971 rather than 1981 and by running ORCHIDEE-CAN with the storm damage module between 1971 and 1980 to adjust the biomass to chronic wind stress. Chronic wind stress occurred mainly in western Norway. Subsequently, this adjusted spatially explicit description of the biomass distribution was used in the simulations for parameter tuning. Trail-and-error tuning started on January 1st 1981 and continued until December 31st 2000 by forcing the model with the CRU-NCEP reanalysis, to simulate the primary damage caused by storm events including the

1999 Anatol storm.

### 3.3 Critical wind speeds for five tree species

Model implementation and parametrization was tested for all five tree species shared between ForestGALES and ORCHIDEE-CAN (see section 2.3.2) by running ORCHIDEE-CAN for a test pixel. The Fontainebleau forest, which is the closest large forest to the LSCE research institute, was arbitrary selected as the simulation site and ORCHIDEE-CAN was run for 200 years

by cycling over the CRU-NCEP reanalysis data from 1901 to 1930. Subsequently, the canopy structure variables simulated by ORCHIDEE-CAN, including $h$, $D$, $C_w$, $C_d$ and $LAI$, were used as the input data for a stand-alone version of ForestGALES (MathCAD version) to estimate the $CWS_{ov}$ and $CWS_{bk}$. In total four types of $CWSs$, $CWS_{ov}$ and $CWS_{bk}$ for the inner and outer forest, were simulated by both models for the same canopy structure.

### 3.4 Critical wind speeds over Southern Scandinavia

Southern Scandinavia was simulated as a 35 by 35 half-degree pixels grid. For each pixel, four critical wind speeds, i.e., $CWS_{ov}$ and $CWS_{bk}$ for the inner and outer area of the forest were simulated making use of the simulated canopy structures for the different tree species and age classes contained in that pixel. To this aim a simulation over Southern Scandinavia was set-up by using the CRU-NCEP reanalysis for 2010 making use of the parameter values for $G_{adj}$ and $R_f$, as derived during the tuning phase. The model was restarted from the time-point December 31st 2010 of the aforementioned simulation of

Naudts et al. (2016), in which forest management and land use were prescribed from the historical reconstructions presented in McGrath et al. (2016). The vegetation over this area mainly consists of coniferous tree species, i.e. Norway spruce (*Picea abies* (L.) Karst) and Scots pine (*Pinus sylvestris* L.), however species mixture between coniferous tree and broadleaved species such as birch (*Betula pendula* Roth and *B. pubescens* Ehrh.) can be found in this region (Drössler, 2010). The simulated spatial

distribution of critical wind speeds for each tree species thus reflects the effects on critical wind speeds of age class structure, forest management and, to a lesser extent, local climate conditions.

## 3.5  Storm damage over Sweden

The capability of ORCHIDEE-CAN to simulate storm damage was evaluated by comparing the observed and simulated storm damage over Sweden from 1951 to 1980 and from 2001 to 2010. A simulation over Sweden was set-up by using the CRU-NCEP reanalysis over the last half-century making use of the parameter values for $G_{adj}$ and $R_f$ as derived during the tuning phase. The region under study entails a 15 by 15 half-degree pixels grid covering southern Sweden and a section of Norway. The state of the forest in Sweden on December 31st 1940 was described by using the matching year from an existing 250-year long simulation (see also section 3.2). Because the 250-year long simulation did not account for the effects of wind damage, the first 10 years from 1941 to 1950 of the simulation-experiment were intended to reach equilibrium between the vegetation and the mean wind speed. For these 10 years the climate data for 1941 to 1950 were used. Within the study domain, this approaches reduced the standing biomass mainly in western Norway (not shown). Subsequently, the simulation-experiment continued from 1951 to 2010, the period for which damage reports are available (Nilsson et al., 2004; Schlyter et al., 2006; Bengtsson and Nilsson, 2007; Gardiner et al., 2010). The years 1981 to 2000 which were used for parameter tuning were excluded from the evaluation.

## 3.6  Biophysical effects of storm damage in Les Landes

The capability of ORCHIDEE-CAN to simulate the effects of storm damage on the surface albedo in the visible range was evaluated. In January 2009, the storm Klaus passed over Les Landes in France. Simulated albedo and leaf area estimates before and after this storm were compared against observed albedo and leaf area values before and after the storm. Les Landes was selected as a study site because time series for albedo were available for the region (Planque et al., 2017). The region under study is covered by a 6 by 6 half-degree pixels grid and the simulation is driven by the CRU-NCEP reanalysis between 1991 and 2010. The state of the forest on December 31st 1990 was described by using the matching year from the same existing 250-year long simulation (see previous section). The first 10 years of the simulation-experiment were intended to reach equilibrium between the vegetation and the mean wind speed. For these 10 years the climate data for 1991 to 2000 were used. Within the study domain, this approach had very little effect on the standing biomass (not shown). Subsequently, the simulation-experiment continued from 2001 to 2010, the period for which the MODIS albedo product was analysed (Planque et al., 2017). Two ORCHIDEE-CAN simulation pixels, one located at the center of Les Landes forest with storm disturbance and another located at the eastern of Les Landes forest without the storm disturbance, were selected for analysing the bio-physical responses to the storm damage for the summer time during the 10 years simulation.

## 4   Results

### 4.1   Downscaling wind fields

The effect of spatial and temporal aggregation on wind fields can be found by the comparison of CRU-NCEP six hour mean wind speed to the stand level 30 min maximum wind speed. Due to spatial and temporal aggregation, the CRU-NCEP six hour mean wind speed reanalysis consistently underestimates the 30 min maximal wind speeds. Stratification by the Beaufort wind scale revealed a clear relationship between this scale and the mean wind ratio for different Beaufort values (see Fig.2). The low value of 6h CRU-NCEP wind speed in the BWS10 bin may due to the fact that all observations in this bin come from a single location wind category (Figure S3). A $4^{th}$ order polynomial function was fitted through the observed $MWR$ to downscale the CRU-NCEP six hour mean wind speed to a max value of 30 min mean wind speed within a six hour time frame.

$$U_{max} = a_0 U_{CRU-NCEP} + a_1 U^2_{CRU-NCEP} + a_2 U^3_{CRU-NCEP} + a_3 U^4_{CRU-NCEP} \tag{12}$$

Where $U_{max}$ is the max value of 30 min mean wind speed ($ms^{-1}$) within a six hour period. $a_0$ to $a_3$ are regression parameters, which has the following values: $a_0 = -5.299$, $a_1 = 2.051$, $a_2 = -0.191$ and $a_3 = 0.006$.

Given the intended use of this analysis to convert the six hour mean wind speed of the CRU-NCEP reanalysis into a likely wind speed used by ORCHIDEE-CAN at the half hourly time scale, the mean ratios of the maximum wind speeds where extracted for each Beaufort wind scale for a six hour averaging period. The value of $MWR$ for a six hour averaging period increased from 1.0 to 6.0 when the Beaufort wind scale went up from scale 1 to scale 11.

### 4.2   Critical wind speeds for five tree species

Given that ForestGALES has been extensively validated (Hale et al., 2015) and that the wind throw module in ORCHIDEE-CAN closely followed the physical processes and empirical formulations from ForestGALES, ORCHIDEE-CAN is expected to simulate similar critical wind speeds as estimated by ForestGALES. This is indeed the case for the five tree species shared the species parameters between ForestGALES and ORCHIDEE-CAN. Figure 3 shows all critical wind speeds (CWSs) simulated by ORCHIDEE-CAN and estimated by ForestGALES with the same canopy structure information, which was closely matched. The difference of CWSs from two models (RMSE for all CWSs) was limited in a few meter per second ranging from 0.4 to 2.0 for different tree species. Moreover, the estimates show that the critical wind speed close to a forest edge is always lower than the respective value further away from a forest edge. (e.g., Fig.3(F) < Fig.3(A), ..., Fig.3(J) < Fig.3(E)).

Also note that for oak, for example, the critical wind speeds for breakage and overturning are almost identical for small trees, but the difference between both critical wind speeds increases with increasing tree height (Fig.3(E) and Fig.3(J)). This implies that taller oak stands are more vulnerable to tree overturning than to stem breakage compared to smaller stands. The relationship between tree height and critical wind speed for spruce is very different compared to oak. For spruce the critical wind speeds for breakage and overturning are within several $ms^{-1}$ from each other and this difference remains more or less constant with increasing tree height (Fig.3(A) and Fig.3(F)). In other words, both stem breakage and tree overturning may occur

simultaneously in tall spruce stands. Furthermore, small beech stands are more vulnerable to breakage (similar to spruce) but tall beech stands are more vulnerable to turnover (similar to oak) (Fig.3(C) and Fig.3(H)).

## 4.3 Critical wind speed over southern Scandinavia

The modelled critical wind speeds for inner and outer forest areas was shown in Fig 4 and these CWSs were compared with daily maximum wind speed. The contour lines in Fig. 4(C)&4(D) indicate the spatial distribution of the difference between CWSs and daily maximum wind speed. Across southern Scandinavia, most of the Norway spruce forests can cope with wind speeds, exceeding 25 $ms^{-1}$ for the outer area (away from forest edge) (Fig. 4(A)), but this minimum critical wind speeds can be reduced around 5 $ms^{-1}$ for the inner area (in a forest edge) (Fig. 4(B)). Stands with old spruces of 25 m tall growing on a well drained soil were simulated to even sustain wind speeds exceeding 40 $ms^{-1}$. The proximity of forest edges is expected to decrease the critical wind speeds (Eqs.(1),(2) and (6)). Indeed, a spruce forest close to a forest edge can sustain wind speeds ranging from 10 to 35 $ms^{-1}$. As already shown in Figs.3(A)&(F), the difference in critical wind speeds between overturning and breakage are very small for spruce.This behaviour is sustained over a large spatial domain (compare Figs.4(A)&4(B)). We further compare the difference between these CWSs the maximum daily wind speed 9th January, 2005. Forests located in the central of the southern Sweden are expected to suffer from storm damage for the Gudrun case (see Fig. 4(C)&4(D))

## 4.4 Storm damage over Sweden

The frequency of storm events in the period from 1981 to 2000 used for parameter tuning is about one storm every five to ten years. The observed primary damage of these storms ranges between 2 to 5 $\times 10^6 \, m^3$ of wood (black line in Fig. 5) . The sensitivity of the simulated damage for different values of the relaxation parameter $R_f$ is presented in Fig.5 (grey lines). Using the prior setting, i.e., $G_{adj} = 1$ and $R_f = 1$) results in an error (RMSE, Hyndman and Koehler (2006)) of about $2 \times 10^6 \, m^3$ for a calibration period without major storms. Higher parameter values, e.g., $G_{adj} = 3.0$, make the model overestimate storm damage up to 10 $\times 10^6 \, m^3$. Parameter tuning ($G_{adj} = 2.45$ and $R_f = 6.0$) resulted in a modest reduction of the deviation between observations and simulations, i.e., a RMSE of $1.35 \times 10^6 \, m^3$, and was therefore used to evaluate the wind throw model.

The RMSE was largely determined by the simulated damage for 1986, 1987 and 2000. For these years, no damage above the background volume was observed despite the presence of high wind speeds in the CRU-NCEP reanalysis over South Sweden in December and January for those three years. Irrespective of the value used for $R_f$, the high wind speeds in the CRU-NCEP reanalysis resulted in substantial damage volumes compared to the observations. This result suggests that ORCHIDEE-CAN would benefit more from improving the representation of gustiness $G_{adj}$ than from refining the damage relaxation parameter $R_f$.

The forest extent has expanded strongly in southern Sweden during the 20th century. In 1940 the forestry system was different from that of today. The clear-felling system that produces the forest edges that are being modelled in this study was not widely adopted until the 1950s. This means that the state of the forest, and the exposure to strong wind, was very different in the beginning and at the end of the simulation period. ORCHIDEE-CAN partly simulating this transition: in 1940 the

average simulated above-ground biomass was $154\,m^3ha^{-1}$ where in 1990 the biomass had increased to $168\,m^3ha^{-1}$. Within this half century the simulated forest area in Sweden increased from $242{,}500\,km^2$ to $247{,}100\,km^2$ and the share of high stand management increased from $54.8\,\%$ to $69.2\,\%$. Despite ORCHIDEE-CAN accounting for the observed forest transition, the model mostly underestimated the damaged wood volumes of large storms.

In January 2005, Gudrun caused the biggest recorded storm damage in Swedish history. We extracted the simulation result from the evaluation experiment and calculated the total simulated storm damage by Gudrun. Simulated storm damage was $76.6 \times 10^6\,m^3$, which is compares well to the reported $75.0 \times 10^6\,m^3$ (Gardiner et al., 2010). Moreover, the high level damage pixels ($> 30\,m^3$ per ha) are located in central Götaland, which demonstrates that, when the driver data locate the storm in the right location, the model has the capability to reproduce the spatial distribution for the event (see Fig. 8).

The model, however, underestimated the storm damage in 2007 (Fig.5). Given the reconstructed wind speed in the driver data, the observed storm damage appears high. The failure of the model to simulate the right order of magnitude in storm damage in 2007 may have two reasons: (1) the CRU-NCEP reconstruction underestimated the wind speed and/or (2) ORCHIDEE-CAN overestimated the critical wind speed by not accounting for the legacy effects of Gudrun such as decreased tree stability owing to root damage or increased gap sizes following Gudrun. Additionally, the observed storm damage is more uncertain
than for Gudrun because the inventory in 2007 was not made as detailed as for Gudrun.

### 4.5 Biophysical effects of storms

In January 2009, Klaus made landfall in Southwesten France nearby Les Landes forest. Following Klaus a decrease in leaf area index and an increase in albedo were reported from SPOT-VGT and MODIS observations, respectively (Planque et al., 2017). Comparison of the simulations against the MODIS observations suggests that the model is able to reproduce the direction of
the changes in the surface characteristic following the passing of Klaus in 2009 (Fig.9(A)-(D)). ORCHIDEE-CAN reproduced both magnitude and direction of the changes in leaf area index (Fig.9(A)). Where the ORCHIDEE-CAN rightly simulated an increase of visible albedo following storm damage (Fig.9(B)), the model overestimated the absolute value of albedo and its change. Although this mismatch could be caused by numerous processes and parameters, the use of a constant PFT-specific background albedo in ORCHIDEE-CAN may limit the capability of the model to correctly simulate the growth of an herb
layer following stand replacing disturbances. In line with meteorological theory, large scale forest disturbances resulted in a decrease in roughness length and transpiration Fig.9(C)-(D)). Decreasing roughness and transpiration could, however, not be confirmed at the site level due to a lack of observations.

## 5    Discussion

### 5.1 Downscaling wind fields

Wind is a highly heterogeneous turbulent flow of air. The flow consists of various sized eddies, and because the energy is conserved, changes in the distribution of the eddy size will result in different flow regimes. Gusts are generated when the flow

passes over a heterogeneous surface such as valleys, ridges, objects or any surface warmer or cooler than its surroundings (Lanquaye-Opoku and Mitchell, 2005; Liu and Weng, 2009). The temporal resolution of a gust is seconds to minutes and its spatial resolution is meters. The highest wind speeds are caused by these gust and are responsible for most of the storm damage (Usbeck et al., 2010).

The success of a large-scale model in simulating storm damage will thus to a large extent depend on the capability of the model to simulate gusts. At present, Large Eddy Simulations are the most advanced approach to simulate the turbulent flow of air (Moeng, 1984; Dupont and Brunet, 2008). The high computational demand of the method (Yang, 2015) currently prevents it from being used in large scale models such as ORCHIDEE-CAN. Likewise, advances in regional atmospheric modelling resulted in the capacity to simulate gusts and gust gradients in the wind field such as tornado (Ishihara et al., 2011) and

hurricanes (Vickery et al., 2009). The computational requirements of regional models is also too high to consider them for global simulations. High computational needs can be avoided by using statistical downscaling but these methods come at the expense of a poor process representation (Larsén and Mann, 2006; Salameh et al., 2009; Huang et al., 2015; Winstral et al., 2017). Moreover, the statistical relationship used for downscaling the gridded wind field from a coarse to a fine resolution depends on the gridded wind field and its spatial and temporal resolutions.

In this study, we used a statistical approach that builds on the relationship between in-situ observations and the gridded reanalysis data. The simplicity of the approach enabled us to focus on implementing, developing and validating the storm damage model rather than improving the physical representation of gusts in ORCHIDEE-CAN. Downscaling the gridded wind field from a coarse resolution to a fine resolution requires accounting for both spatial and temporal aggregation. Our approach made use of the mean wind ratio, to convert the mean wind speed from six hour CRU-NCEP reanalysis into a mean wind speed

at the 30 min time step. By further analysing the contribution of spatial and temporal aggregation on downscaling (see Figure S3 in SI), it was found that the temporal averaging was responsible for about 40% of the reduction of the variation whereas spatial averaging was responsible for the remaining 60%. In other words, simulating storm damage in large scale models would initially benefit most from increasing the spatial resolution as that would allow to better account for the extreme wind speeds.

Note, that in ORCHIDEE-CAN (revision 4262) the downscaled wind speed is only used to test whether the critical wind

speed is exceeded and if so, to simulate the resulting damage from stem-break and overturning. Consequently, the downscaled wind speeds are not used for calculating surface roughness, momentum, heat exchange and water vapour transfer. The current statistical downscaling is thus not suitable for use in model set-ups where ORCHIDEE-CAN is coupled to an atmospheric circulation model, for example, LMDz (Hourdin et al., 2006). Simulating storm damage from coupled land-atmosphere models, e.g., LMDz / ORCHIDEE-CAN, will thus require an effort to better represent turbulent air flow in the planetary boundary layer.

**5.2   Storm damage over Sweden**

The state of the forest in Sweden on December 31st 1940 was described by using the matching year from an existing simulation, which ran from 1750 to 2010 (Naudts et al., 2016). This existing simulation accounted for land cover changes and changes in forest management following the historical reconstruction of forest cover and management by McGrath et al. (2015), but it did

not account for storm damage. However, the forest reconstruction map shows that the forest species over Sweden are mainly consisted of Scots pine, Norway spruce and birch which is comparable with the report made by Helmfrid (1991).

Given its strength, the 1999 storm Anatol resulted in relatively little damage ($5 \times 10^6\,m^3$), likely because it has hit the southernmost part of Sweden where the landscape contains fewer forests and the share of broad-leaf forest is much higher than just a few tens of kilometres north of the storm track. Although the good match between the data and the model (Fig. 5) is due to the fact that Anatol occurred within the tuning period, the conditions that are held responsible for the low volume of storm damage were indeed found to be reproduced. The CRU-NCEP reanalysis suggests the Anatol storm track hitting southern Sweden, and ORCHIDEE-CAN uses a forest index of 56 % in the storm track, but 68 % for landscapes 100 km north of the track. Furthermore, in 1999, 67 % of the simulated forests were broad-leaved in southern Sweden whereas roughly one third were 100 km north of the track. Finally, ORCHIDEE-CAN simulates higher critical wind speeds for broad-leaved trees (especially in winter) than for conifers (Fig.4)

Simulated storm damage between 1951 and 2010, excluding the years 1981 to 2000 which were used for parameter tuning, was used to evaluate the model. Within this period, major wind storms occurred in 1954, 1969 and 2005 and were associated with observed storm damage ranging from 20 to $75 \times 10^6\,m^3$. Note that the storm damage reported for the evaluation events is about ten times higher than the damage reported for the events used to parametrize the model. The model error is $5.05 \times 10^6\,m^3$ in the evaluation period. When the large storms in 1954, 1969 and 2005 are excluded in the calculation of the RMSE, the model errors decreases to $1.62 \times 10^6\,m^3$ (see Fig.6). The RMSE is thus largely due to underestimating the damage during big storms.

## 5.3 Subpixel heterogeneity

Gusts and thus storm damage are often generated by surface heterogeneities, for example, forest edges, surface topography, and landscape features with a very different temperature than the surrounding landscape. Where these heterogeneities can be accounted for in Large Eddy Simulations and regional models, large scale models such as ORCHIDEE-CAN managed to limit their computational costs by, among other simplifications, ignoring these subpixel heterogeneities (Krinner et al., 2005). When implementing processes that are partly driven by these heterogeneities, i.e. storm damage, these models are thus operated at the limit of their design. In this study, we have tried to overcome this issue by reconstructing subpixel heterogeneity from the wood harvest aggregated over the last 5 years and assuming that all gaps are square-shaped and have a surface area of 2 ha (see Eqs.7-8). This approach enabled ORCHIDEE-CAN to calculate separate critical wind speeds for forest close to a forest edge (inner) and forests away from a forest gap (outer).

Although this approach is thought to have contributed to reproducing the observations, it lacks at least three well documented subpixel heterogeneities: (1) in ORCHIDEE-CAN the number of gaps varies over time but their surface area is set constant. Hence, individual gaps do not increase in size and the relationship between gap size and gustiness (Peltola, 2006) is not accounted for; (2) Although each pixel has an altitude, elevation differences within a pixel are not considered. Thus, ORCHIDEE-CAN does not account for the effects of exposure on tree species distribution (Ruel et al., 1997); and (3) In ORCHIDEE-CAN, all tree species within a pixel share the same water column, hence the model does not account for the

interactions between tree species and long-term soil water content heterogeneity (Ringeval et al., 2012). All three omissions may have contributed to the failure of the model to simulate the storm damage in 2007 (see Fig.6).

It is expected that simulating gap size and gap dynamics would improve the performance of the storm damage model. Performance improvements, however, would equally apply to simulating fires because the size, age and composition of subpixel

forest patches were identified as important elements in simulating fire risk and fire behaviour (Keane et al., 2004). Further improvements can be expected for simulating forest regeneration because gap size will determine the growing environment for the recruitment (Rüger et al., 2009). It has even been suggested that to reduce the uncertainty in terrestrial vegetation responses to future climate and atmospheric $CO_2$, a change in research priorities away from biomass production and toward structural dynamics and demographic processes should be favoured (Friend et al., 2014). Simulating subpixel heterogeneity is emerging

as the next frontier in large-scale models, however, the challenge to do so at low computational costs remains.

## 6   Conclusions

The representation of storm damage in ESMs could use empirical models building upon the relationships between total storm damage and predictor variables such as topography, vegetation, soil, historical forest management maps, and wind exposure (Scott and Mitchell, 2005; Kamimura et al., 2008; Lagergren et al., 2012; Moore et al., 2013; Takano et al., 2016). As an

alternative, that better suits the process-based philosophy of Earth System Modelling, more mechanistic approaches could be used. A mechanistic approach, however, requires information on the canopy structure (Peltola et al., 1999; Gardiner et al., 2000). This requirement rules out including mechanistic wind damage models in large-scale land surface models because the latter make use of the big-leaf assumption to represent the canopy. Recently, the big-leaf assumption was replaced by a three dimensional representation of the canopy structure in the large-scale land surface model ORCHIDEE-CAN (Naudts et al.,

2015). This change in canopy representation enabled incorporating the process formalizations of the stand-level wind risk model called ForestGALES (Gardiner et al., 2000) into ORCHIDEE-CAN.

Incorporating ForestGALES into ORCHIDEE required solving three issues related to differences in spatial scales between both models. The spatial resolution of ForestGALES is $10^3 \, m^2$ whereas the typical resolution used in applications with large-scale land surface models such as ORCHIDEE-CAN is $10^9 \, m^2$. (1) This difference in spatial scales implies that subpixel

heterogeneity had to be accounted for in ORCHIDEE. Although not all sources of subpixel heterogeneity were simulated, vegetation structure at the landscape level was accounted for (see section 2.2.3). (2) The spatial scale of ORCHIDEE required that wind fields had to be downscaled to account for the occurence of gusts. A statistical downsacling approach was used in this study (see section 2.3.1). Model performance could, however, benefit from a more mechanistic approach to downscale the wind fields to better account for gusts, irrespective of whether the wind fields come from gridded reanalysis data or simulations from

an atmospheric circulation model coupled to ORCHIDEE-CAN. A more mechanistic approach to downscale the wind fields may require accounting for subpixel heterogeneity, but it is also needed to enable the use of the storm damage functionality in tandem with atmospheric models, such as the WRF model (Stéfanon et al., 2012) or LMDZ (Hourdin et al., 2006). (3) When ForestGALES is run at a small spatial scale it is fair to assume that all trees will be damaged when the critical wind speed

for that stand is exceeded. This assumption no longer holds for the spatial scales used in ORCHIDEE-CAN. The model was, therefore, completed by an empirical function to convert the difference between the wind speed and the critical wind speeds into forest damage.

The storm damage functionality of ORCHIDEE-CAN was then parametrized and tested for selected case studies in southern Sweden and south-western France. The model largely captured the 60-year long-term storm damage dynamics over the Swedish forests and simulated a decrease in leaf area and an increase in visible albedo following storm damage in France. The model was thus shown to have the flexibility to reproduce diverse observations, although the validity of the parameters outside Sweden and France still needs to be demonstrated.

In the long-term, building the capacity to simulate the impact of wind storms will result in a better understanding of the climate response to the biotic and abiotic disturbance agents in the Earth system, but as well support actionable science to evaluate the effects of changing storm frequency and intensity on global food production, and the effectiveness of forest management in mitigating and adapting to climate change.

## 7 Code availability

The ORCHIDEE-CAN code and the LibIGCM run environment are open source and distributed under the CeCILL licence (http://www.cecill.info/index.en.html). The ORCHIDEE-CAN branch is available via the follow web link (https://forge.ipsl. jussieu.fr/orchidee/browser/branches/ORCHIDEE-DOFOCO/ORCHIDEE?rev=4262 or https://doi.org/10.5281/zenodo.1109750). Readers interested in running ORCHIDEE-CAN (revision 4262) are encouraged to contact the corresponding author for full details and latest bug fixes.

*Author contributions.* FP and YC implemented ForestGALES into ORCHIDEE-CAN. YC and SL designed the study and tested the wind throw model. KB and BG provided species specific and other parameters for the wind throw model and in-situ observations of storm damages over Sweden. CP provided the SPOT-VGT and MODIS satellite observations for France. JR, AV, JO, KN and MJM developed and tested ORCHIDEE-CAN revision 2566 to which the wind throw model was added. YC set up and processed all model simulations, and all authors contributed to writing the manuscript.

*Acknowledgements.* This work was funded through a bilateral agreement between the Swedish Research Council (VR) and the French Alternative Energies and Atomic Energy Commission (LSCE-IPSL, CEA, CNRS and UVSQ). Part of computational resources was provided by the Research Center for Environmental Changes (RCEC) at Academia Sinica (RCEC, AS) and National Central University (NCU) in Taiwan. YC received funding through Ministry of Science and Technology, R.O.C. (MOST 106-2111-M-001-001-MY3). SL was in part funded through an Amsterdam Academic Alliance fellowship.

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

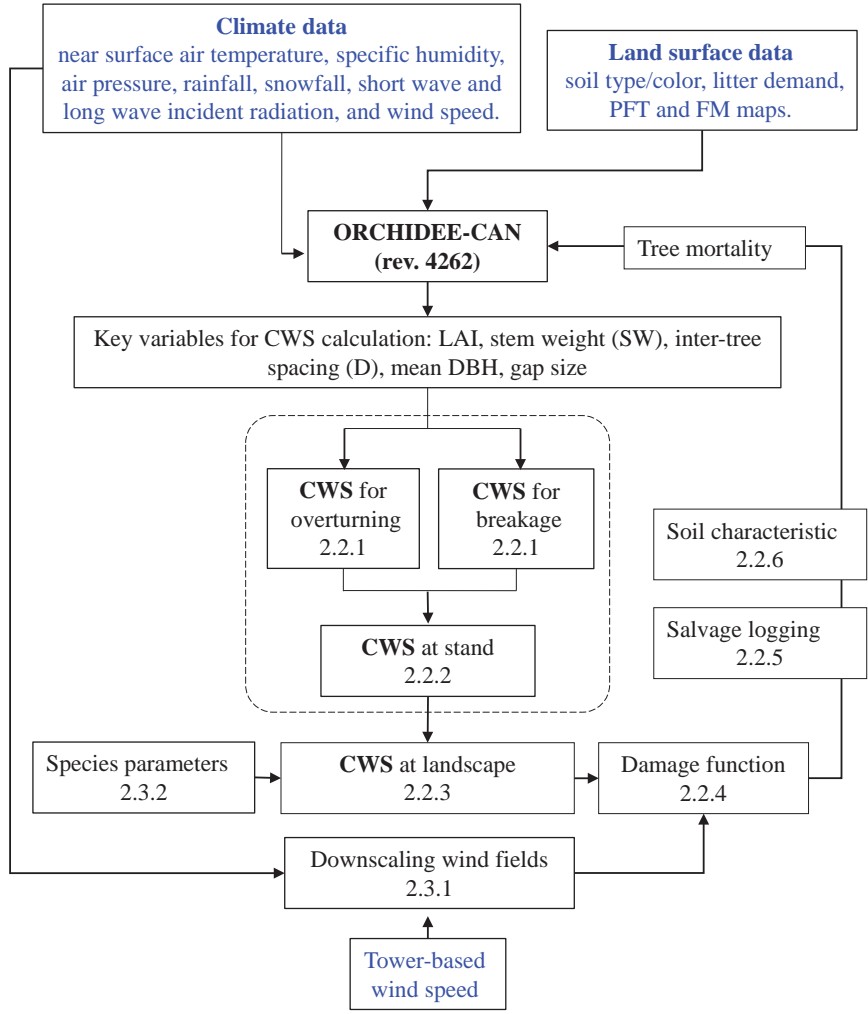

**Figure 1.** Information flow of this study showing the link between the different elements presented in sections 2.2 and 2.3 (numbers in the figure refer to section numbers in the text). The diagram shows input data in blue color. The dashed box shows critical wind speeds calculations according to ForestGALES.

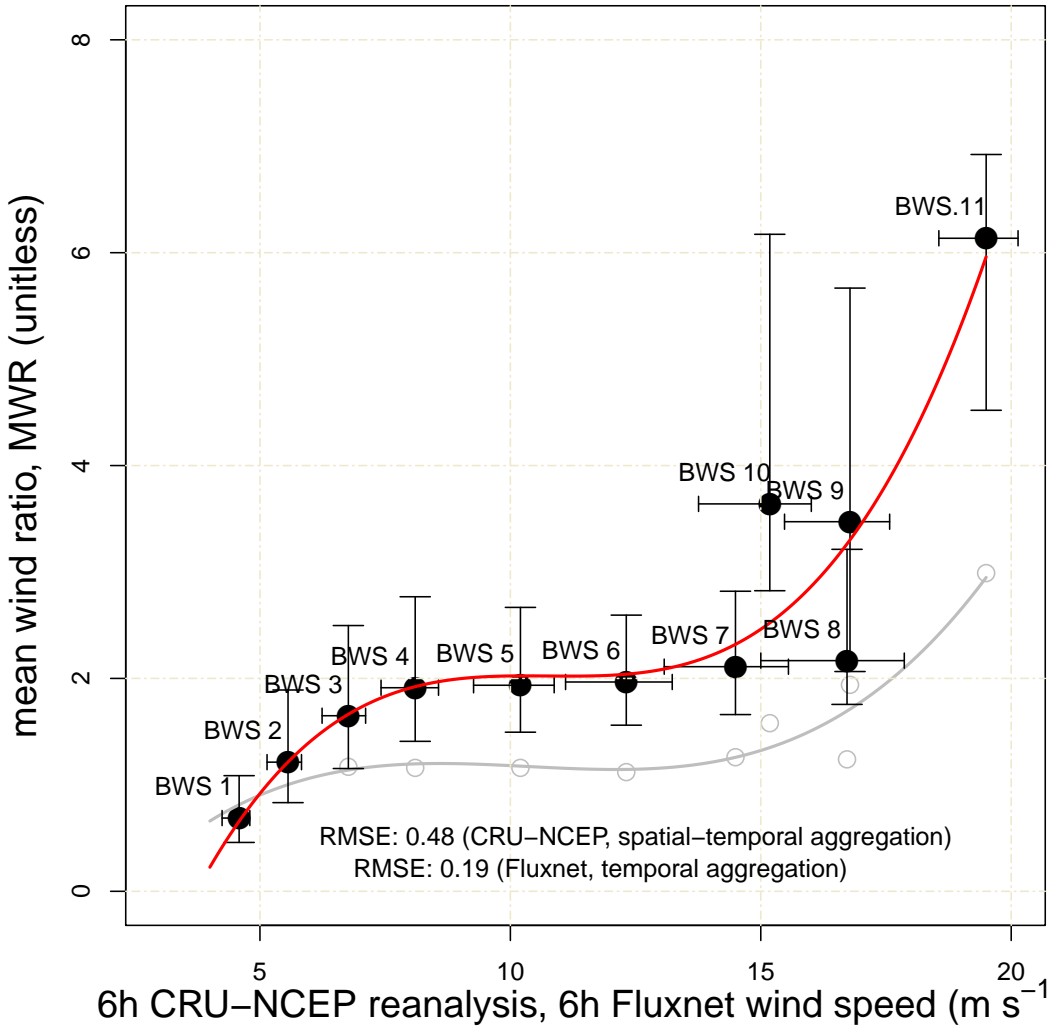

**Figure 2.** Distribution of the mean wind ratio (MWR) in each Beaufort wind scale (BWS) and the relationship between the six hour CRU-NCEP reanalysis wind speed and MWR. Fitting of the relationship (red line) used Eq.(12) with regression coefficients $a_0 = -5.299$, $a_1 = 2.051$, $a_2 = -0.191$ and $a_3 = 0.006$. This relationship is used to convert CRU-NCEP six hour mean wind speed to the 30 min maximum wind speed in this study. The RMSE of using this regression model to predict the mean value of MWR in each BWS class is 0.48. The open circles (gay color) show the effect of 6h temporal aggregation on the MWR from the selected Fluxnet sites in European forest. The gray line is the fitting line of the open circles.

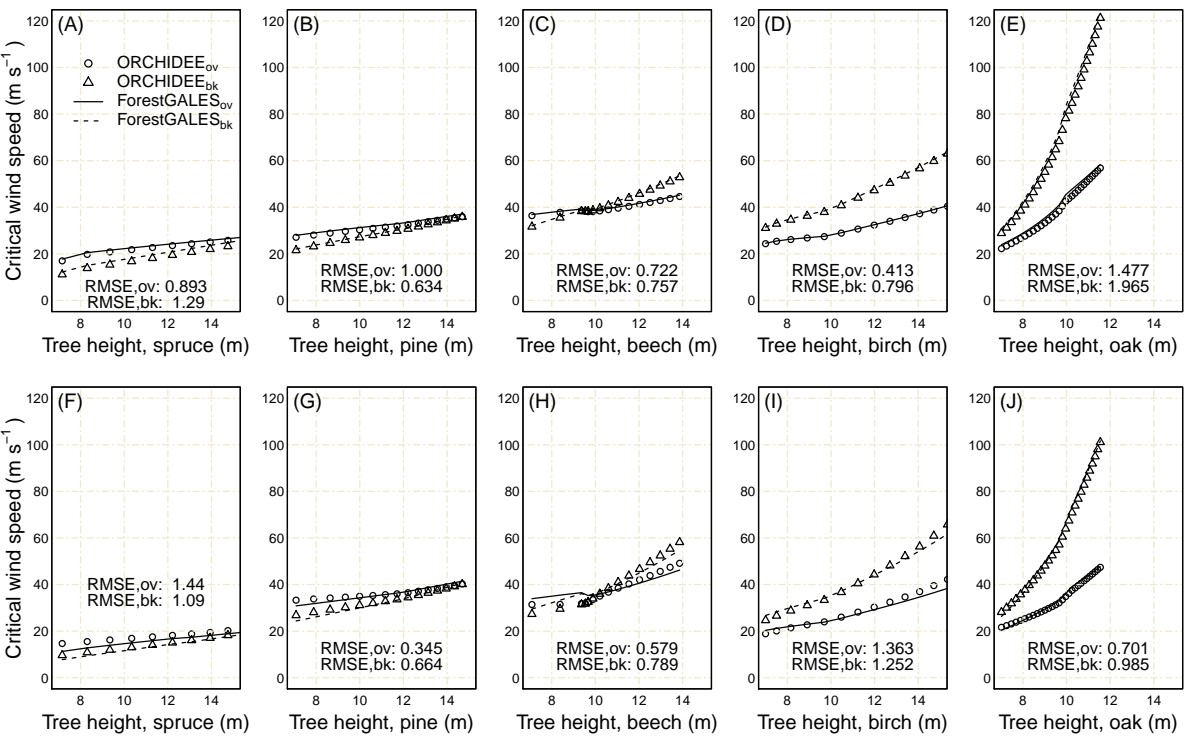

**Figure 3.** Model calculated critical wind speeds as a function of tree height for five common tree species in Europe. For each tree species the critical wind speed is calculated for overturning and stem breakage for forest located away from a forest edge (outer, upper panels A-E) and nearby a forest edge (inner, lower panels F-J). The ORCHIDEE-CAN simulations are shown as symbols and benchmarked against the ForestGALES simulations which are shown as lines. The scientific names of the tree species are given in section 2.2.3. The CWSs difference between the ORCHIDEE-CAN and ForestGALSE calculation were calculated using Eq.11, which the CWSs from ForestGALSE were treated as the reference.

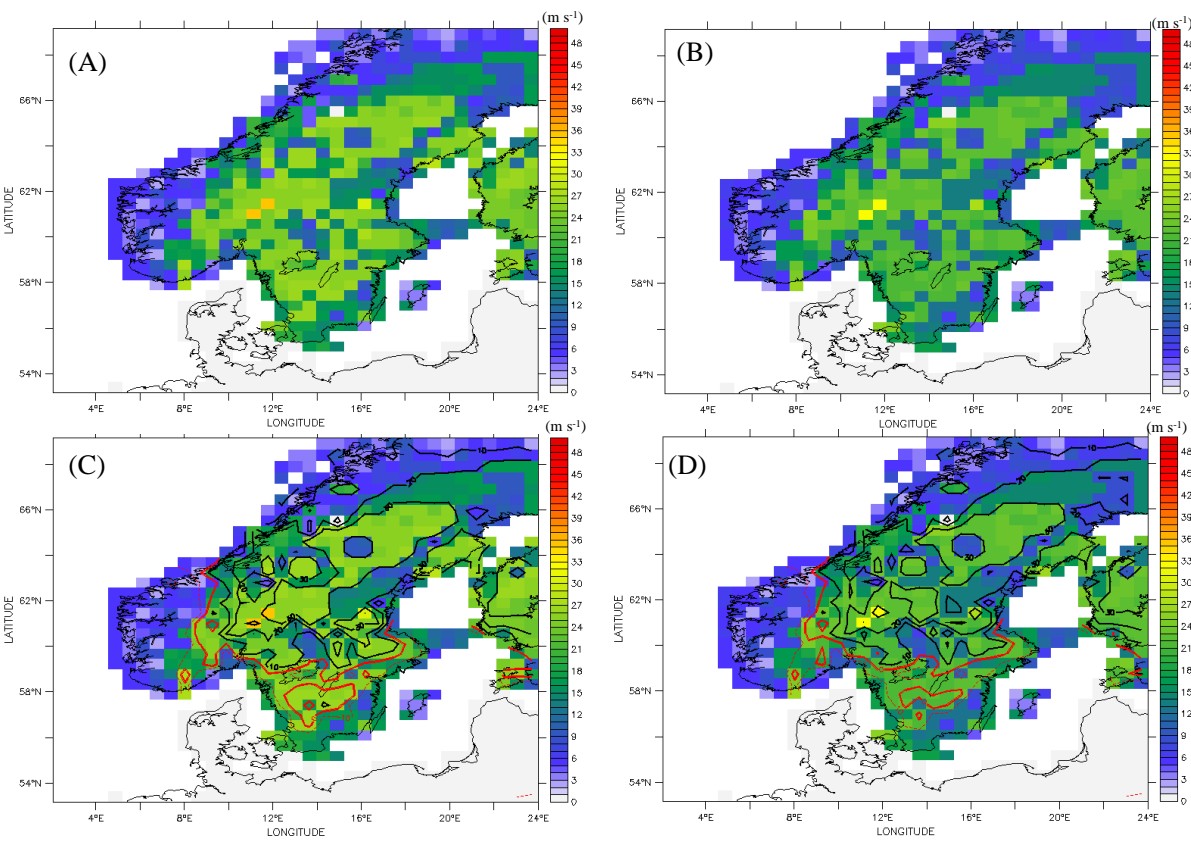

**Figure 4.** The ORCHIDEE-CAN calculated lowest critical wind speeds for overturning or stem breakage for forest located near to (inner) and away (outer) from a forest edge. When making the display, the critical wind speeds from the three diameter classes and four age groups from $Picea$ species were assessed and the lowest value was compared against the daily maximum wind speed for estimating the damage due to storms. Lowest critical wind speeds in the forest away from a forest edge (outer) (A), lowest critical wind speeds in the forest near to a forest edge (inner) (B), lowest critical wind speeds overlaid with the difference between maximum daily wind speed and lowest critical wind speed in outer area on 9th January, 2005 (C), lowest critical wind speeds overlaid with the difference between maximum daily wind speed and lowest critical wind speed in the outer area (D). The contours show the positive wind speed difference in black and the negative wind speed difference in red. Forests within the red contours are expected to suffer from storm damage.

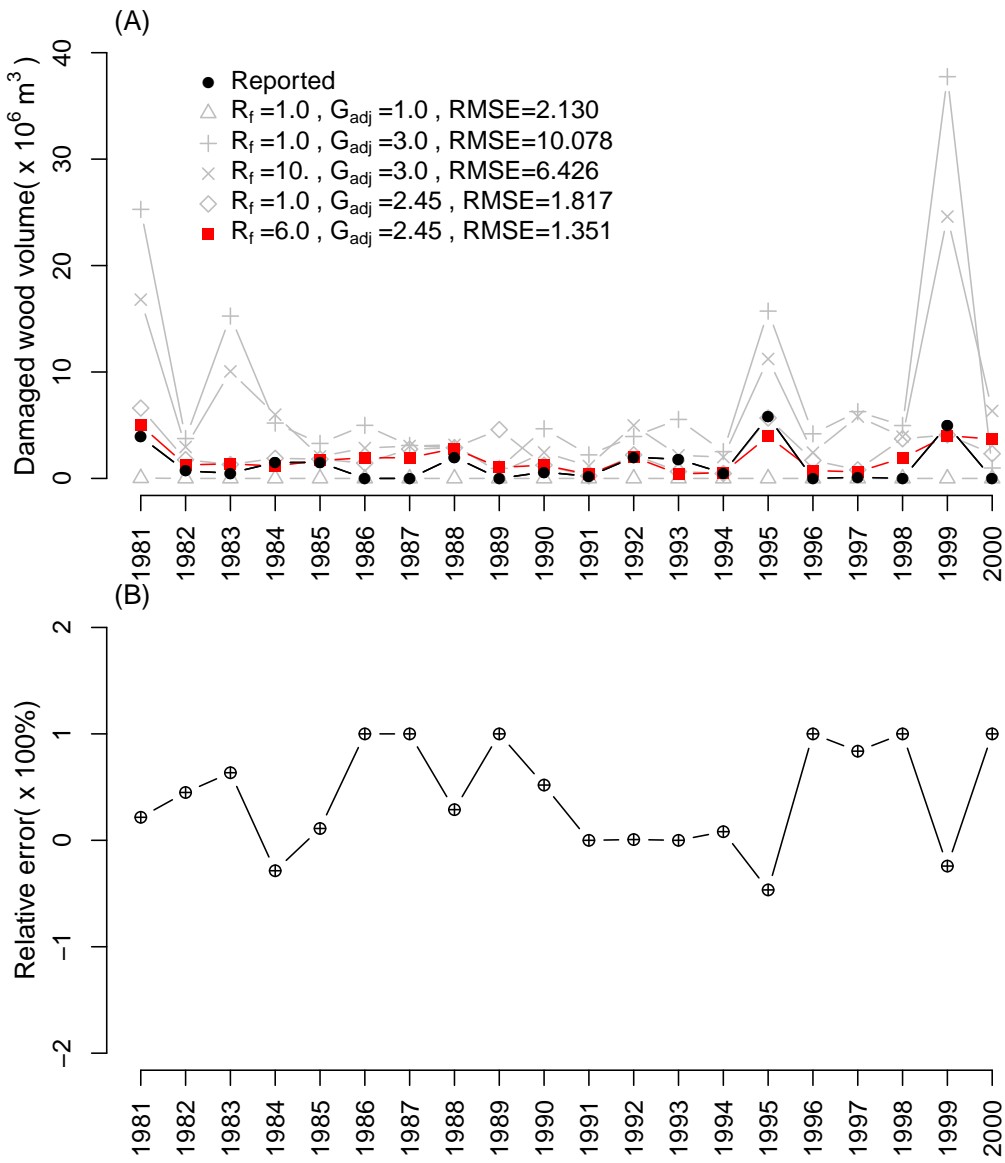

**Figure 5.** Sensitivity of the simulated storm damage over Sweden between 1981 and 2000 for different values for the relaxation parameter ($R_f$) and the gust factor adjustment $G_{adj}$ (A). Observed storm damage is extracted from (Nilsson et al., 2004; Schlyter et al., 2006; Bengtsson and Nilsson, 2007; Gardiner et al., 2010). The relative model simulation error (($estimation - observation$)/$estimation$) for the best tuned case (B).

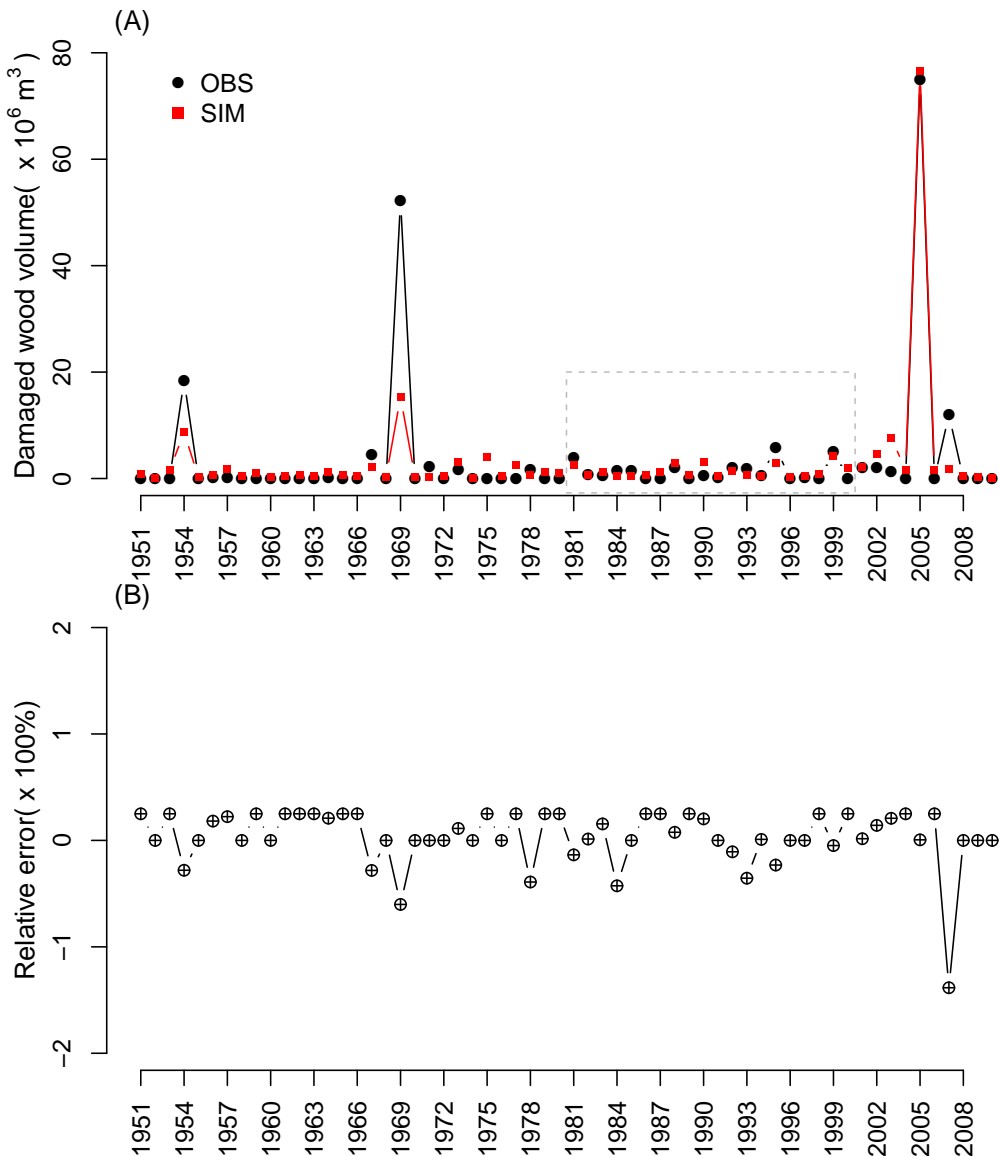

**Figure 6.** Comparison of the storm damage simulated by the ORCHIDEE-CAN and the reported wood damage over the Sweden from 1951 to 2010. Observed storm damage is extracted from (Nilsson et al., 2004; Schlyter et al., 2006; Bengtsson and Nilsson, 2007; Gardiner et al., 2010) The dashed-line area is the period from 1981 to 2000, which was selected for parametrization. The RMSE of the estimated storm damage is $1.35 \times 10^6 \, m^3$ for the parametrization period and $5.05 \times 10^6 \, m^3$ during the evaluation period. The validation period ranges from 1951 to 2010 but excludes the years 1981 to 2000. The relative model simulation error for the validation period from 1981 to 2000 (B).

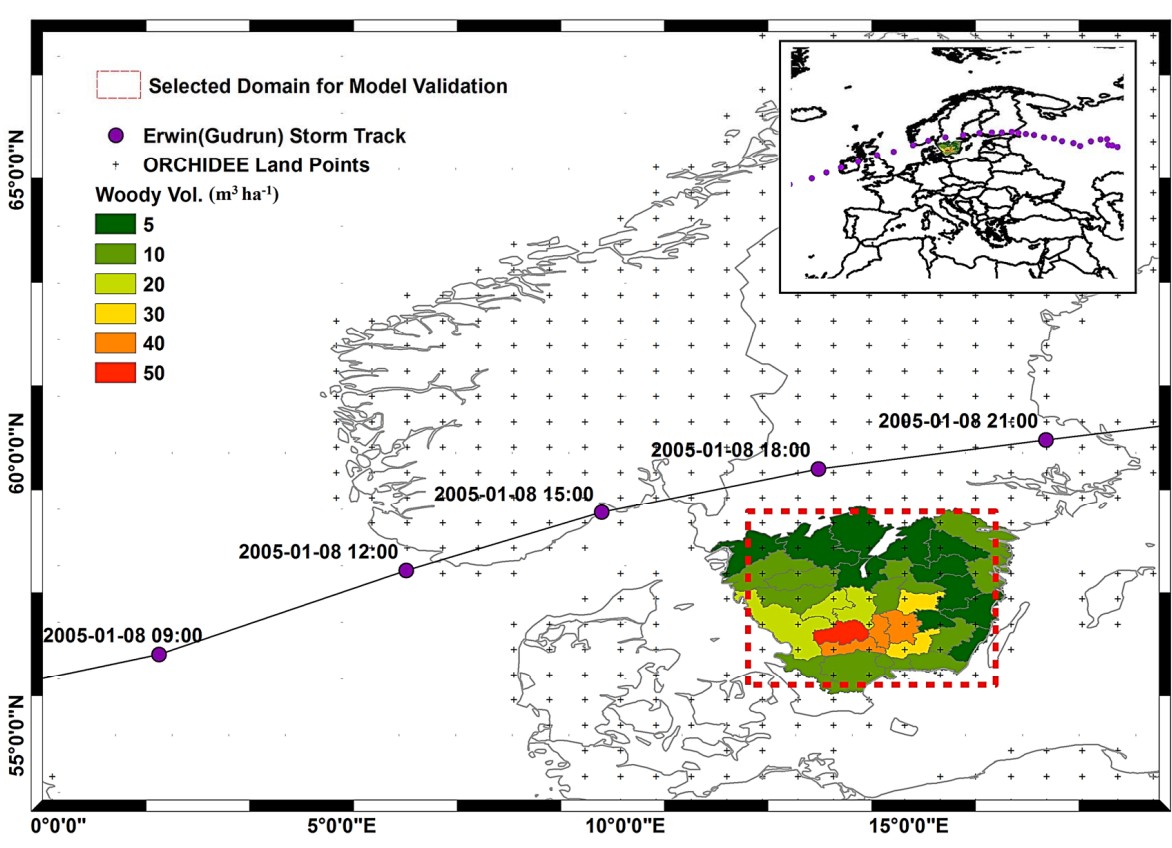

**Figure 7.** The case study of the storm Edwin(Gudrun) on 8th Jan 2005. The color scale shows the damaged woody volume ($m^3 ha^{-1}$) due to this event.

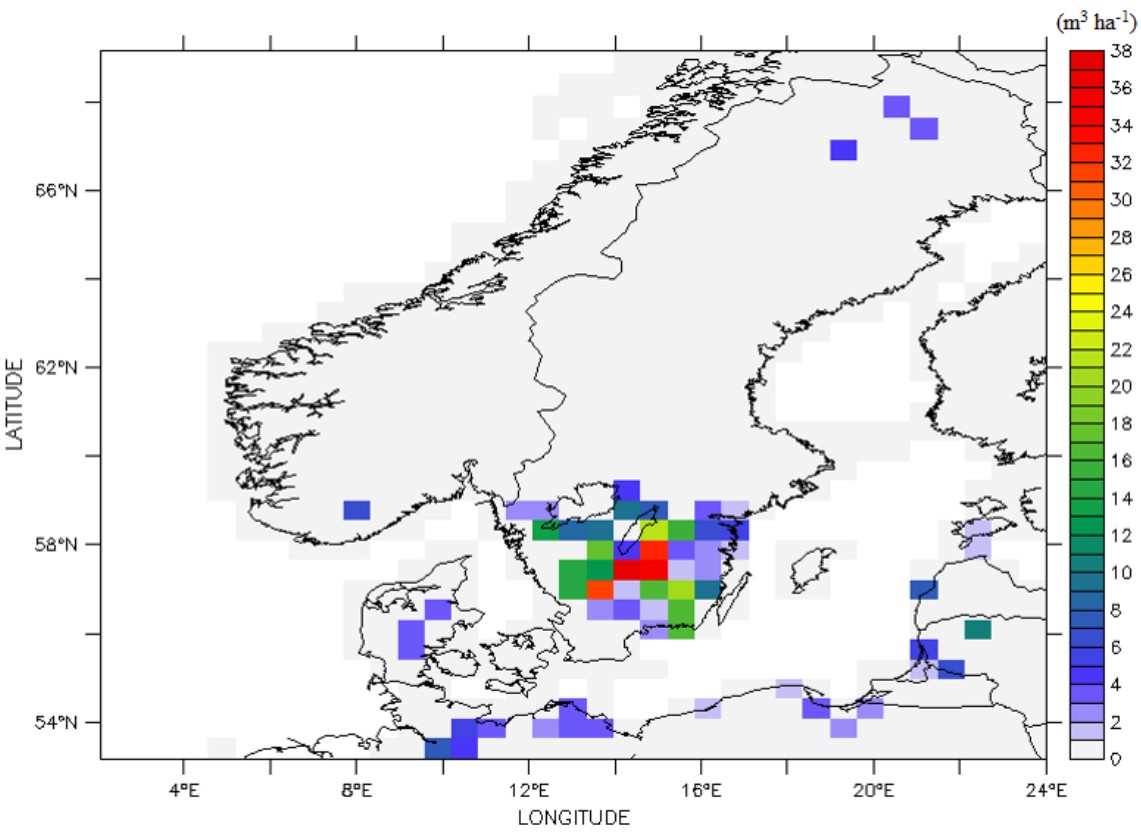

**Figure 8.** Simulated spatial distribution of the damaged wood volume($m^3 ha^{-1}$) over southern Sweden by Gudrun in January 2005.

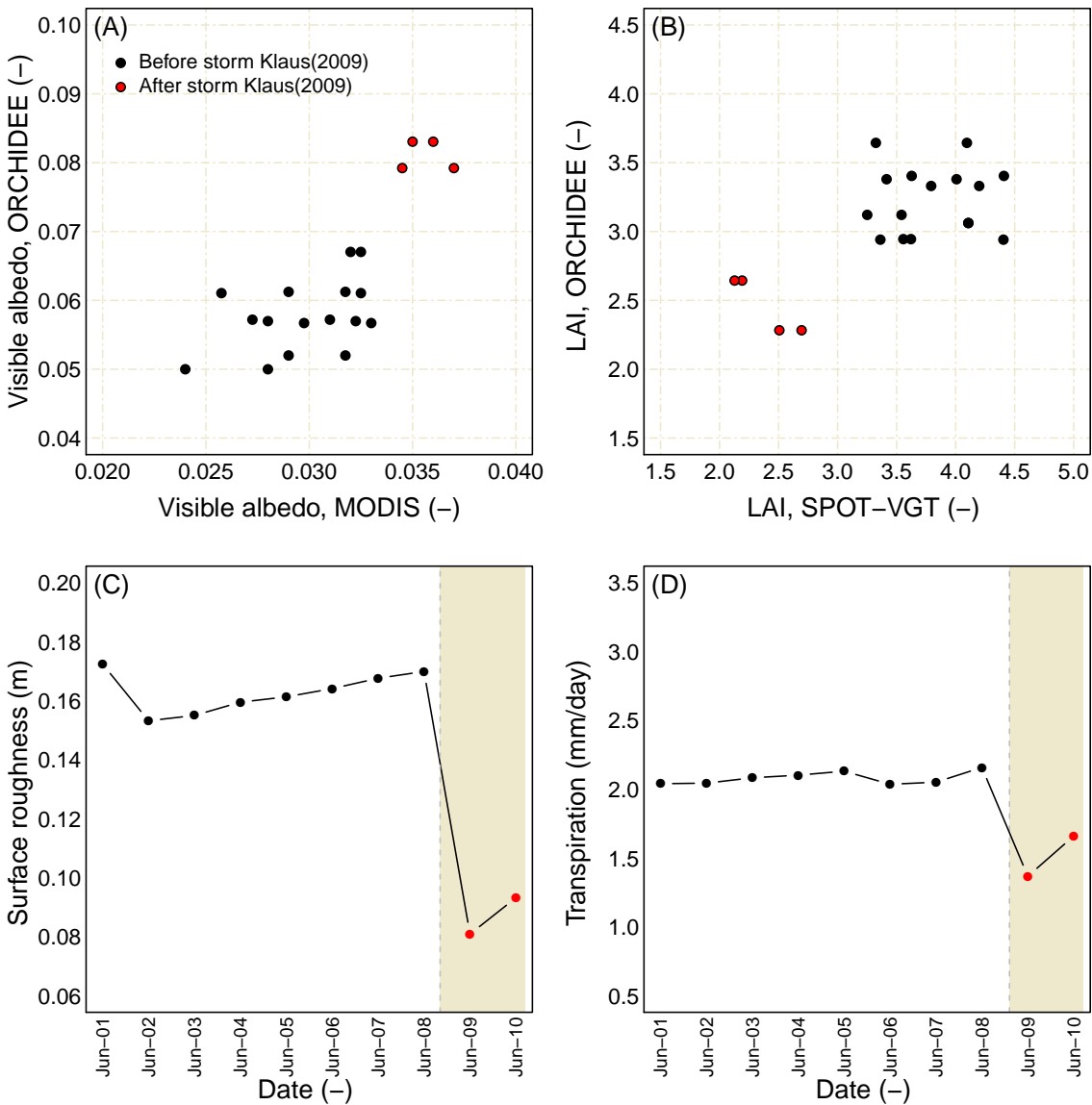

**Figure 9.** Effects of the wind storm Klaus (January 2009) on the forest in Les Landes, France. Comparison of the ORCHIDEE-CAN simulated visible albedo and the MODIS observations in summertime (June) from 2001 to 2010 (A). Comparison of the ORCHIDEE-CAN simulated leaf area index (LAI) and SPOT-VGT derived estimates in summertime (June) from 2001 to 2010 (B). The dynamics of roughness for the most damage pixel in Les Landes forest simulated by ORCHIDEE-CAN at monthly time scale from 2001 to 2010 (C). The summertime (June) transpiration rate for the most damaged pixel in Les Landes forest simulated by ORCHIDEE-CAN (D). The shading area indicates the period after the passing of Klaus in January, 2009.

**Table 1.** Symbolic notation used throughout the paper.

| Symbol | Description | Unit | Symbol in the module |
|---|---|---|---|
| $a_{0,\dots,3}$ | Regression parameters for nonlinear fitting | unitless | a_0 - a_3 |
| $A_5$ | Forest area of timber removals in previous five years | $m^2$ | area_timber_removals_5_years |
| $A_{gap}$ | Gaps area within a modelled grid | $m^2$ | area_gap |
| $A_{grid}$ | Modelled grid area | $m^2$ | area_icir |
| $A_{inner}$ | Forest area around the gaps | $m^2$ | area_total_close |
| $A_{outer}$ | Forest area far away the gaps | $m^2$ | area_total_further |
| $c_{d1}$ | A regression constant for surface roughness | unitless | cd1 |
| $C$ | Drag coefficient scale parameter | unitless | streamlining_c |
| $C_d$ | Length of the live crown | m | canopy_depth |
| $C_{reg}$ | Regression between stem weight (SW) and resistance to overturning | $N\,m\,kg^{-1}$ | overturning_moment_multiplier |
| $C_w$ | Maximum width of canopy | m | canopy_breadth |
| $C_D$ | Drag coefficient | unitless | porosity_sub |
| $CWS_{ov}$ | Critical wind speed for tree overturning | $m\,s^{-1}$ | cws_overturn |
| $CWS_{bk}$ | Critical wind speed for stem breakage | $m\,s^{-1}$ | cws_break |
| $d$ | Zero-plane displacement | m | d_wind |
| $dbh$ | Stem diameter at breast height (at 1.3 m) | m | mean_dbh |
| $D$ | Average spacing between trees | m | current_spacing |
| $D_\beta$ | Damage rate to the forest stands | unitless | wind_damage_rate |
| $D_{\max}$ | A maximum damage rate to the forest stands | unitless | max_damage |
| $f_{CW}$ | Dimensionless factor to account for additional turning moment due to crown and stem weight | unitless | f_crown_mass |
| $f_{edge}$ | Dimensionless factor to account for the tree position relative to the edge | unitless | edge_factor |
| $f_{knot}$ | Dimensionless factor to account for reduction in clear wood MOR due to knots | unitless | f_knot |
| $G$ | Dimensionless factor to account for gustiness of wind | unitless | gust_factor |
| $G_{adj}$ | A linear parameter for adjusting the gustiness of wind | unitless | s_factor |
| $\gamma$ | A regression function for canopy structure parametrisation | unitless | gamma_solved |
| $h$ | Tree height | m | mean_height |
| $\kappa$ | von Karman constant | unitless | ct_karman |
| $LAI$ | Leaf area index at level | $m^2\,m^{-2}$ | lai |
| $MOR$ | Modulus of rupture on wood for species of interest | Pa | modulus_rupture |
| $MWR$ | Mean wind ratio from | unitless | max_wind_ratio |
| $n$ | Parameter controlling reduction in drag coefficient with wind speed | unitless | streamlining_n |
| $\Psi_h$ | A correction function for atmospheric stability | unitless | psih_sub |
| $\rho$ | Density of air | $kg\,m^{-3}$ | air_density |
| $R_f$ | A relaxation parameter to adjust the damage rate | unitless | sfactor |
| $SW$ | Stem (bole) weight | kg | stem_mass |
| $u_h$ | 30 min mean wind speed at the canopy height | $m\,s^{-1}$ | uh_speed |
| $U_{max}$ | Maximum value of 30 min mean wind speed within a 6h period | $m\,s^{-1}$ | u_daily_max |
| $U_{Fluxnet}$ | Maximum value of mean wind speed from Fluxnet every 12 samples | $m\,s^{-1}$ | u_fluxnet |
| $U_{CUR-NCEP}$ | Mean wind speed from CRU-NCEP reanalysis dataset | $m\,s^{-1}$ | u_cruncep |
| $x$ | Distance from forest edge | m | tree_heights_from_edge |
| $z_0$ | Surface roughness | m | z0_wind |

**Table 2.** Parameter values used in the ORCHIDEE-CAN windthrow module. The scientific names of the tree species are given in section 2.3.2.

| Parameters | Unit | Spruce | Pine | Beech | Birch | Oak | Rooting depth |
|---|---|---|---|---|---|---|---|
| Regression factor, $C_{reg}$ | $(Nmkg^{-1})$ | 125.6 | 139.6 | 198.5 | 128.8 | 198.5 | Shallow |
|  |  | 146.6 | 162.9 | 198.5 | 128.8 | 198.5 | Deep |
| Knot correction factor, $f_{knot}$ | (unitless) | 0.9 | 0.85 | 1.0 | 1.0 | 1.0 |  |
| Overhanging correction factor, $f_{CW}$ | (unitless) | 1.136 | 1.136 | 1.136 | 1.136 | 1.136 |  |
| Green density | $(kg/m^3)$ | 960 | 1020 | 1030 | 930 | 1060 |  |
| Streamlining factor, n | (unitless) | 0.51 | 0.75 | 0.90 | 0.88 | 0.85 |  |
| Streamlining factor, C | (unitless) | 2.35 | 3.07 | 2.41 | 1.96 | 2.66 |  |
| Module of rupture, MOR | $(Pa)$ | 3.6E+7 | 4.6E+7 | 6.5E+7 | 6.3E+7 | 5.9E+7 |  |
| Maximum damage rate, $D_{max}$ | (unitless) | 1.0 | 1.0 | 1.0 | 1.0 | 1.0 |  |
| Gustiness adjust factor, $G_{adj}$ | (unitless) | 2.45 | 2.45 | 2.45 | 2.45 | 2.45 |  |
| Harvest ratio | (%) | 99.0 | 99.0 | 99.0 | 99.0 | 99.0 |  |
| Relaxation factor, $R_f$ | (unitless) | 6.0 | 6.0 | 6.0 | 6.0 | 6.0 |  |