# Peer review of "Simulating damage for wind storms in the land surface model ORCHIDEE-CAN (revision 4262)"

_Geoscientific Model Development, 2017_

## Referee Comment (RC1) · Anonymous Referee #1 · 23 Oct 2017

Simulating damage for wind storms in the land surface model ORCHIDEE-CAN (revision 4262) Yi-Ying Chen1, Barry Gardiner, Ferenc Pasztor, Kristina Blennow, James Ryder, Aude Valade, Kim Naudts, Juliane Otto , Matthew J. McGrath, Carole Planque, and Sebastiaan Luyssaert

General Comment The authors incorporate a well-established wind disturbance model (ForestGALES) into a dynamic global vegetation model, the ORCHIDEE-CAN. It is perhaps the first study of windthrows simulation by an Earth System Model (ESM). I emphasize the novelty of this study because it improves our understanding of an overlooked agent of tree mortality (wind) in forest ecosystems.

Comments/Concerns: 1. Winds are a major agent of tree mortality, a well-known fact that has been discussed extensively in the literature over a range of spatial scales and ecosystems. Yet, the introduction of this study is very limited and does not justify why windthrows need their own representation scheme in an ESM. Furthermore, there is not a formal definition of wind storms. Wind storms can vary from strong winds to tropical cyclones. The frequency and the spatial scales of these events justify this study. However the reader is left to wonder whether this type of study is important. 2. The use of a sigmoid function to represent the storm damage (Equation 9) was not justified. 3. The calculation of critical wind speed uses five tree species. How representative are these species of the whole simulation domain? Are these tree from the same Genus or Family? 4. The comparison of modeled damage versus the observational data would benefit from the inclusion of percentage. 5. Critical wind speed and downscaling require more detail. What is the wind speed needed to overturn trees in the study area? How does it compares with the critical wind speed? 6. An analysis of forest damage focusing on pixel heterogeneity vs wind speed is relevant for this study.

---

## Referee Comment (RC2) · Anonymous Referee #2 · 23 Oct 2017

The article entitles, "Simulating damage for wind storms in the land surface model ORCHIDESS-CAN" is aimed to develop an earth system model using the sub-model, ORCHIDESS-CAN, with focusing on storm wind damage in forests. In terms of wind damage, a mechanistic-empirical model, ForestGALES, was coupled with ORCHIDESS-CAN. I understand that developing such models is really important to understand the environmental system and to make strategies for climate change. However, I think the scientific originality in this article seems weak partly due to the writing style. When using existed models in research, it is very important to show what the new ideas and findings are. At this moment, this article is not well organized. The structure of the method section need to be improved. The result section is hard to understand

and some paragraphs should be located in Discussion. The results are not well discussed in Discussion. In addition, comparison between calculated and observed data requires some statistical analysis in order to show how much reliable what you did. Some ambiguous expressions were found in Results. Please find my comments as followed and I hope you will improve your article.

1. The section of Methods should be improved. I think the structure and components need to be modified. For example, first models used in the article are explained including the parameters and equations. Second, what your original ideas are explained. A figure (diagram or flowchart) would be helpful to show the process. Third, validation procedures can be explained including the input data of three regions with references. Also please explain why you chose the regions and different analysis were conducted for them.

2. Please explain what ORCHIDESS-CAN can do and how to use the model with required inputs. Is ORCHIDESS-CAN a grid-based model? Can we change the grid size (resolution)? Has the structural growth model been included in the model? How did you exactly integrate ORCHIDESS-CAN and ForestGALES? Did you make new codes? I think a diagram would be helpful to understand how the models work and what you actually did in the study.

3. I suppose that you used some observed data such as satellite photos and forest damage data. But some of the data sources were not indicated in Methods.

4. In terms of the equations, I do not understand why 9h needs to be doubled in eq. 1. How did you calculate the average mean tree height and distance between trees? Are they averaged in a single grid? What data was used to calculate them?

5. In this article a term, actual wind speed, is quite often found. However, I do not think we can obtain actual wind speed data except maybe around an anemometer. The wind speeds used here would be "estimated" wind speeds.

[Figure]

6. In results, some methods and terms such as spatio-temporal comparison, Beaufort wind scale, and root-mean-square error are not explained in Methods.

7. Please try to write the results as simple as possible. Some sentences and paragraphs seem to be unnecessary or better to move to Discussion. For instance, I do not think the first sentence on page 12 is important in Results. The second and fourth paragraphs on page 13 should be located in Discussion.

8. In Figure 1, the mean wind ratios of BWSs 8, 9, 10, and 11 should be explained. Although a fitting line was created, I am not sure how it is reliable. For example, if BWS 11 or BMS 9 are outliners, the line will become different. Why are wind speeds in BWS 10 lower than those in BWSs 8 and 9? To use the fitting line for the following analysis, it is important to justify how much statistically suitable the line is.

9. In terms of Figure 2, are there comparisons between the critical wind speeds calculated by ForestGALES only and those by ORCHIDESS-CAN + ForestGAELS? If so, are the differences only from input data to the models? In addition, to compare the speeds between stem breakage and uprooting, it is important to conduct some statistical analysis in order to show how statistically different or similar between the two outputs are. Which graphs are for the forest edges?

10. In Figure 3, I think the lowest critical wind speeds are more important to consider wind damage rather than the averaged critical wind speeds. Also, are these results for the current forest conditions?

11. In terms of the results from Les Landes, I do not understand why these indicators were chosen to represent the results although they seem to be interesting. It is important to clearly address in Methods what kind of analysis would be conducted in advance. Why 20 values were used in (A) and (B) in Figure 8? Where these values came from?

12. Please discuss your results in Discussion. Some of the discussions are found in

Methods, so it is better to move the parts to the Discussion section. I am not sure whether the first paragraphs are necessary especially at the beginning of Discussion. For example, gusts are not directly analyzed in this research. Is 6.3 Salvage logging really important for your research, although there are no indications in the results? I think in the first parts of Discussion, using the same order of Results would help readers to easily understand your discussions.

---

## Short Comment (SC1) · 6 Nov 2017

As explained in https://www.geoscientific-model-development.net/about/manuscript_types.html GMD is is encouraging authors to provide a persistent access to the released source code. It is suggested that authors upload the program code of models (including relevant data sets) as a supplement or make the code and data available at a data repository preferable with an associated DOI (digital object identifier) for the exact model version described in the paper.

Many thanks. Lutz Gross GMD Executive Editor

---

## Author Comment (AC1) · 20 Nov 2017

Reply to the Referee #1 for the manuscript (gmd-2017-174) submitted to GMD.

We would like to thank the referee for the comments on the manuscript. In this document we discuss the concerns of the referee and indicate how we could improve the presentation of this study if we are invited to submit a revised manuscript.

**Responses to the Referee#1 : General Comment**

The authors incorporate a well-established wind disturbance model (ForestGALES) into a dynamic global vegetation model, the ORCHIDEE-CAN. It is perhaps the first study of windthrows simulation by an Earth System Model (ESM). I emphasize the novelty of this study because it improves our understanding of an overlooked agent of tree mortality (wind) in forest ecosystems.

Thank you for your kind remark.

**Responses to the Referee#1 : Specific concerns**

(1) Winds are a major agent of tree mortality, a well-known fact that has been discussed extensively in the literature over a range of spatial scales and ecosystems. Yet, the introduction of this study is very limited and does not justify why windthrows need their own representation scheme in an ESM. Furthermore, there is not a formal definition of wind storms. Wind storms can vary from strong winds to tropical cyclones. The frequency and the spatial scales of these events justify this study. However the reader is left to wonder whether this type of study is important.

We agree that wind storms are not properly defined in the manuscript and we will add the definition of wind storms in the revised manuscript. We feel that the importance of simulating wind storm damage is rather extensively addressed in the discussion (Wind storm disturbances and their climate feedbacks). We will move this section forward and present it in the introduction)

(2) The use of a sigmoid function to represent the storm damage (Equation 9) was not justified.

Although different damage function with different dependencies could be conceived, for example, soil moisture dependent, topographic dependent and wind speed dependent, we considered that the wind speed variation is the most important factor in controlling the storm damage in a forest stand. A wind speed dependent approach was used and validated by Anyomi et al. (2017) for various

tree species in the temperate climate. We will refer to this paper as justification for using this type of wind damage function to estimate the storm damage rate in our revised manuscript.

Reference: Anyomi, K. A., Mitchell, S. J., Perera, A. H., and Ruel, J. C.: Windthrow dynamics in Boreal Ontario: A simulation of the vulnerability of several stand types across a range of wind speeds, *Forests*, 8, 1–15, doi:10.3390/f8070233, 2017.

(3) The calculation of critical wind speed uses five tree species. How representative are these species of the whole simulation domain? Are these tree from the same Genus or Family?

Spruce, pine and birch make up almost the entire forest cover of Southern Sweden. Pine is the single most important species in Les Landes. These regions were only used to test the model. The simulation domain of this new development is Europe. The five species for which the model was tested make up 67% of the European forest cover. In terms of taxonomic families the representativeness increases to >90% (Novel Maps for Forest Tree Species in Europe, Renate Köble and Günther Seufert).

Reference: Köble R. and Seufert G.: Novel maps for forest tree species in Europe. Proceedings of the 8th European Symposium on the Physico-Chemical Behaviour of Air Pollutants: "A Changing Atmosphere!", Torino (It) 17-20 September 2001.

(4) The comparison of modeled damage versus the observational data would benefit from the inclusion of percentage.

We agree and will add the relative model simulation errors ((estimation-observation)/estimation) to the figures for the comparison of modeled damage versus the observational data. The new figures would then like the example shown below.

[Figure]

Fig 1. Sensitivity of the simulated storm damage over Sweden between 1981 and 2000 for different values for the relaxation parameter ($R_f$) and the gust factor adjustment $G_{adj}$ (A). Observed storm damage is extracted from (Nilsson et al., 2004; Schlyter et al., 2006; Bengtsson and Nilsson, 2007; Gardiner et al., 2010). The relative model simulation error ((estimation-observation)=estimation) for the best tuned case (B).

[Figure]

Fig 2. Comparison of the storm damage simulated by the ORCHIDEE-CAN and the annual primary wood damage over the Sweden from 1951 to 2010. Observed storm damage is extracted from (Nilsson et al., 2004; Schlyter et al., 2006; Bengtsson and Nilsson, 2007; Gardiner et al., 2010) The dashed-line area is the period from 1981 to 2000, which was selected for parametrization. The RMSE of the estimated storm damage is $1.35 \times 10^6 m^3$ for the parametrization period and $5.05 \times 10^6 m^3$ during the evaluation period. The validation period ranges from 1951 to 2010 but excludes the years 1981 to 2000. The relative model simulation error for the validation period from 1981 to 2010 (B).

(5) Critical wind speed and downscaling require more detail. What is the wind speed needed to overturn trees in the study area? How does it compares with the critical wind speed?

We propose to increase the level of detail in the revised manuscript by showing the modelled critical wind speeds for inner and outer forest areas and compared these CWS with daily maximum wind speed. We then used the contour lines to show the spatial distribution of the difference between minimum CWS and daily maximum wind speed (Fig. 4 in our revised manuscript). The new figure would then look like the example below.

[Figure]

Fig 3. The ORCHIDEE-CAN calculated lowest critical wind speeds for overturning or stem breakage for forest located near to (inner) and away (outer) from a forest edge. When making the display, the critical wind speeds from the three diameter classes and four age groups from *Picea* species were compared. Note that this lowest value is used and compared with the daily maximum wind speed for estimating the damage due to storms. Lowest critical wind speeds in the forest away from a forest edge (outer) (A), lowest critical wind speeds in the forest near to a forest edge (inner) (B), lowest critical wind speeds overlaid with the difference between maximum daily wind speed and lowest critical wind speed in outer area on 9th January, 2005 (C), lowest critical wind speeds overlaid with the difference between maximum daily wind speed and lowest critical wind speed in the outer area (D). The contours show the positive wind speed difference in black and the negative wind speed difference in red. Forests within the red contours are expected to suffer from storm damage.

(6) An analysis of forest damage focusing on pixel heterogeneity vs wind speed is relevant for this study.

It is unclear to us what the referee had in mind when making this comment. Nevertheless, we realize that the current implementation overlooks many sources of sub-pixel heterogeneity when calculating storm damage but we would like to stress that one of the novelties of this study is that we found a solution (i.e., the calculation of an inner and outer area) to use the stand-level GALES at a much larger scale while avoiding the need to increase the spatial resolution (and thus the computational costs) of ORCHIDEE. Following a comment by the other referee, this novelty will be stressed in the methods, discussion and conclusion. Further improvements towards accounting for sub-pixel heterogeneity are the topic of ongoing funding applications.

---

## Author Comment (AC2) · 20 Nov 2017

Reply to the Referee #2 for the manuscript (gmd-2017-174) submitted to GMD.

We would like to thank the referee for the comments on the manuscript. In this document we discuss the concerns of the referee and indicate how we could improve the presentation of this study if we are invited to submit a revised manuscript.

Responses to the Referee#2 : General Comment

The article entitles, "Simulating damage for wind storms in the land surface model ORCHIDESS-CAN" is aimed to develop an earth system model using the submodel, ORCHIDESS-CAN, with focusing on storm wind damage in forests. In terms of wind damage, a mechanistic-empirical model, ForestGALES, was coupled with ORCHIDESS-CAN. I understand that developing such models is really important to understand the environmental system and to make strategies for climate change. However, I think the scientific originality in this article seems weak partly due to the writing style. At this moment, this article is not well organized. The structure of the method section need to be improved. The result section is hard to understand and some paragraphs should be located in Discussion. The results are not well discussed in Discussion. In addition, comparison between calculated and observed data requires some statistical analysis in order to show how much reliable what you did. Some ambiguous expressions were found in Results.

When invited to submit a revised manuscript, we will pay attention to the structure of the manuscript. Nevertheless, we would like to stress that given the aim of GMD, we decided to focus this manuscript on presenting how ORCHIDEE was further developed to accommodate the principles of GALES. The new version of ORCHIDEE was tested (reported in the results) but given the scope of the journal the effects of wind damage over the test regions were not discussed in any detail. If we would have had access to other data, we could have picked any other region in the world to test the model. In other words, we have no special interest in storm damage Sweden or France. It just happened that due to our professional network we had access to good validation data for these regions. Therefore, it was decided to focus the discussion on the model philosophy and its possible improvements. In the future we would like to use ORCHIDEE+GALES to simulate the storm damage at a large scale, not just Southern Sweden and Southwestern France.

When using existed models in research, it is very important to show what the new ideas and findings are.

We agree with the referee that this aspect is insufficiently developed in the manuscript. In the revised version of the manuscript we will stress the following

novelties:

- Almost the same group of authors developed the model capability to simulate forest and canopy structure. Although these developments are already published they are essential for the work presented here.
- In this manuscript we suggest a numerically cheap solution to deal with one of the key aspects of sub-pixel heterogeneity, i.e., recent forest edges.
- In this manuscript we went beyond the capabilities of GALES by adding and parameterizing a damage function.
- In this manuscript we suggest a solution to account for aggregated driver data which are typically used in large scale simulations.

These novelties will be listed in the abstract and the discussion. In the methods section we will better indicate whether the presented method comes from GALES or was developed as part of this study.

**Responses to the Referee#1 : Specific concerns**

(1) The section of Methods should be improved. I think the structure and components need to be modified. For example, first models used in the article are explained including the parameters and equations. Second, what your original ideas are explained. A figure (diagram or flowchart) would be helpful to show the process. Third, validation procedures can be explained including the input data of three regions with references. Also please explain why you chose the regions and different analysis were conducted for them.

If we correctly understand the reviewer, the suggestion is to merge the information currently presented in section 2 (models) and 3 (parameters). This can be done. As mentioned above we will indicate which developments are new and which were taken from GALES. A flowchart linking all elements presented in the methods could look like the figure shown below.

We are confused by the suggestion to make a third section that presents the validation procedures and the available input data for the test cases as we believe that is exactly what is done in section 4. We revising the manuscript we will look for opportunities to better structure the information currently contained in sections 2, 3 and 4.

[Figure]

Fig 1. Information flow of this study showing the link between the different elements presented in sections 2 and 3 (NOTE: the numbering will be adjusted in the revised manuscript as section 2 and 3 will be merged). The diagram shows input data in blue color. The dashed box shows how the critical wind speeds were calculated in GALES. These critical wind speeds were then compared against critical wind speeds from ORCHIDEE.

(2)  Please explain what ORCHIDESS-CAN can do and how to use the model with required inputs. Is ORCHIDESS-CAN a grid-based model? Can we change the grid size (resolution)? Has the structural growth model been included in the model? How did you exactly integrate ORCHIDESS-CAN and ForestGALES? Did you make new codes? I think a diagram would be helpful to understand how the models work and what you actually did in the study.

In section 2 we will add a description of the basic principles of ORCHIDEE. This description is based on the default description as shown on the web site of the model. When adding these details to the manuscript, parts of the text will have to be rewritten to avoid duplication with the current text and to improve the text flow.

"ORCHIDEE is the land surface model of the IPSL (Institute Pierre Simon Laplace) Earth System Model. Hence, by conception, the ORCHIDEE model can be run coupled to a global circulation model. In a coupled set-up, the atmospheric conditions affect the land surface and the land surface, in turn, affects the atmospheric conditions. However, when a study focuses on changes in the land surface rather than on the interaction with climate, ORCHIDEE can be run off line as a stand-alone land surface model. The stand-alone configuration receives the atmospheric conditions such as temperature, humidity and wind, to mention a few, from the so-called "forcing files". Unlike the coupled set-up, which needs to run at the global scale (but with the possibility of a regional zoom), the stand alone configuration can cover any area ranging from the global domain to a single grid point.

Given its use as the land surface model of the IPSL, ORCHIDEE simulates the terrestrial water and the energy balance, biogeochemical processes such as the carbon cycle, and anthropogenic activities. The biophysical process include latent, sensible, and kinetic energy exchanges at the surface of soils. Heat dissipation and water fluxes are vertically distributed in the soil and the runoff is collected in rivers and lakes. The simulated processes that affect the global carbon cycle include photosynthesis, carbon allocation, litter decomposition, soil carbon decomposition, and maintenance and growth respiration and vegetation dynamics. The anthropogenic activities and more specifically forest management and its effects on the biophysical and biogeochemical processes were developed in ORCHIDEE-CAN (Bellassen et al 2011 and Naudts et al 2015).

Although ORCHIDEE-CAN does not enforce a spatial or temporal resolution, the model does use a spatial grid and equidistant time steps. The spatial resolution is an implicit user setting that is determined by the coarsest resolution of the forcing data and the boundary conditions, i.e., the vegetation distribution, climatological forcing data, and the soil map. If higher resolution drivers are available the model can then be run at that scale. If site-level drivers are available then simulations at the site scale are feasible.

ORCHIDEE-CAN builds on the concept of meta-classes to describe vegetation

distribution. By default it distinguishes 13 such meta-classes (one for bare soil, eight for forests, two for grasslands and two for croplands). Each meta-class can be subdivided in an unlimited number of Plant functional types (PFTs). By default, each meta-class has a single PFT. When simulations make use of species–specific parameters and age classes, several PFTs belonging to a single meta-class will be defined. Biogeochemical and biophysical variables are calculated for each PFT, where most of the biogeochemical variables are reported at the PFT level, the biophysical variables are aggregated at the pixel level because the atmospheric model does not distinguish PFTs and hence its spatial resolution is limited to the pixel scale.

For the water and heat balance of the soil, three soil columns are distinguished: one containing each meta-class that includes forest, one for every meta-class with grass and crops and, finally, one for bare soils. Water and heat related soil variables are calculated separately for each column.

ORCHIDEE-CAN can run on any temporal resolution, however, this apparent flexibility is rather restricted as the processes are formalized at given time steps: half-hourly (i.e. photosynthesis and energy budget), daily (i.e. net primary production) and annual time step (i.e. vegetation dynamics). Hence, meaningful simulations have a temporal resolution of 15 minutes to one hour for the energy balance, water balance and photosynthesis calculations."

Reference:
Bellassen, V., le Maire, G., Guin, O., Dhôte, J., Ciais, P., and Viovy, N.: Modelling forest management within a global vegetation model – Part 2: Model validation from a tree to a continental scale, *Ecol. Model.*, 222, 57–75, 2011.
Naudts, K. et al..: A vertically discretised canopy description for ORCHIDEE (SVN r2290) and the modifications to the energy, water and carbon fluxes, *Geosci. Model Dev.*, 8, 2035–2065, 2015.

(3) I suppose that you used some observed data such as satellite photos and forest damage data. But some of the data sources were not indicated in Methods.

Thanks for pointing out this issue. When preparing the manuscript we indeed overlooked adding a detail description of the observational data. We will add more information about the in-situ observations. This information will be added into an extra section 4.5.

(4) In terms of the equations, I do not understand why 9h needs to be doubled in eq. How did

you calculate the average mean tree height and distance between trees? Are they averaged in a single grid? What data was used to calculate them?

Fig 2 shows the concept of inner and outer area generated by a gap created by human disturbance in a managed forest. Given the suggestions by both reviewers, the revised manuscript will already contain 10 figures, therefore, we prefer not to add this figure as it will be available through this discussion)

[Figure]

Fig 2. Illustration of inner and outer area generated by gaps within a forest area.

(5) In this article a term, actual wind speed, is quite often found. However, I do not think we can obtain actual wind speed data except maybe around an anemometer. The wind speeds used here would be "estimated" wind speeds.

Fair point. Given that also the critical wind speeds are estimated we propose to simply use "wind speed" and "critical wind speed" throughout the manuscript.

(6) In results, some methods and terms such as spatio-temporal comparison, Beaufort wind scale, and root-mean-square error are not explained in Methods.

The reference for the Beaufort wind scale classification and the definition of root-mean-square error will be added into our revised manuscript.

(7) Please try to write the results as simple as possible. Some sentences and paragraphs seem to be unnecessary or better to move to Discussion. For instance, I do not think the first sentence on page 12 is important in Results. The second and fourth paragraphs on page 13 should be located in Discussion.

Although we agree with the referee that part of the material presented in section 5.4 could also be presented in the discussion, we though and still think that it makes a stronger manuscript if the test cases are not promoted to the topic of the discussion. Passing the tests is just a minimum requirement. The test themselves are not the topic of the manuscript. The topic of the manuscript is the model and therefore our discussion focusses on the model.

We consider the first sentence on page 12 to be important because it justifies why we believe that a model-model comparison is, in this case, a valid surrogate for a more traditional model-data comparison.

The changes proposed by the referee as straightforward to implement. Given that the other referee was not disturbed by which material was presented in the results and which material was presented in the discussion, we leave it to the editor to make the final suggestion.

(8) In Figure 1, the mean wind ratios of BWSs 8, 9, 10, and 11 should be explained. Although a fitting line was created, I am not sure how it is reliable. For example, if BWS 11 or BMS 9 are outliners, the line will become different. Why are wind speeds in BWS 10 lower than those in BWSs 8 and 9? To use the fitting line for the following analysis, it is important to justify how much statistically suitable the line is.

We fully agree with this suggestion. The new figure is shown below.

[Figure]

Fig 3. Distribution of the mean wind ratio (MWR) in each Beaufort wind scale (BWS) and the relationship between the six hour CRU-NCEP reanalysis wind speed and MWR. Fitting of the relationship (red line) used Eq.(11) with regression coefficients a0 = -5.299, a1 = 2.051, a2 = -0.191, a3 = 0.006 and RMSE=0.48. This relationship is used to convert CRU-NCEP six hour mean wind speed to the 30 min maximum wind speed in this study.

(9) In terms of Figure 2, are there comparisons between the critical wind speeds calculated by ForestGALES only and those by ORCHIDESS-CAN + ForestGAELS? If so, are the differences only from input data to the models? In addition, to compare the speeds between stem breakage and uprooting, it is important to conduct some statistical analysis in order to show how statistically different or similar between the two outputs are. Which graphs are for the forest edges?

The issue was also pointed out by the referee#1. We will replace the original figure by the new figure3 as shown as below:

[Figure]

Fig 4. The ORCHIDEE-CAN calculated lowest critical wind speeds for overturning or stem breakage for forest located near to (inner) and away (outer) from a forest edge. When making the display, the critical wind speeds from the three diameter classes and four age groups from *Picea* species were compared. Note that this lowest value is used and compared with the daily maximum wind speed for estimating the damage due to storms. Lowest critical wind speeds in the forest away from a forest edge (outer) (A), lowest critical wind speeds in the forest near to a forest edge (inner) (B), lowest critical wind speeds overlaid with the difference between maximum daily wind speed and lowest critical wind speed in outer area on 9th January, 2005 (C), lowest critical wind speeds overlaid with the difference between maximum daily wind speed and lowest critical wind speed in the outer area (D). The contours show the positive wind speed difference in black and the negative wind speed difference in red. Forests within the red contours are expected to suffer from storm damage.

This figure shows the modelled critical wind speeds (CWSs) for inner and outer forest areas and compared these CWSs with daily maximum wind speed. We then used the contour lines to show the spatial distribution of the difference between CWSs and daily maximum wind speed

(10) In Figure 3, I think the lowest critical wind speeds are more important to

consider wind damage rather than the averaged critical wind speeds. Also, are these results for the current forest conditions?

We will replace the current Figure 3 by a new Figure (see above) which only shows the lowest critical wind speed for inner and outer area. This information is then overlaid with the difference between the lowest critical wind speed and the maximum daily wind speed. These results are for the forest condition on 9th January, 2005. Note that the model only uses the lowest critical wind speed (and not the average) to calculate storm damage. As suggested by the referee, the figure was inconsistent with the model code and model approach.

(11) In terms of the results from Les Landes, I do not understand why these indicators were chosen to represent the results although they seem to be interesting. It is important to clearly address in Methods what kind of analysis would be conducted in advance. Why 20 values were used in (A) and (B) in Figure 8? Where these values came from?

We complied the remote sensing data used from the study by Planque et al. (2017) the selected points are shown in pink arrows and the ORCHIDEE simulation grids are overlaid in black lines. We then ran the model for ten years from 2001 to 2010 and extracted the ORCHIDEE outputs from two selected pixels in white box, one near the eastern part of Les Landes forest and another near the middle of Les Landes forest.

[Figure]

Fig 5. Illustration of selected points in the les lands forest.

Although, the temporal and spatial resolution from remote sensing and model simulation are different, we merged these two datasets into ten year summer-time slots in two locations. This is the reason for having 20 points in the scatter plot of comparing the remote sensing observation and model simulation. We will add this part of description into the section 4.5.

Reference:
Planque, C., Carrer, D., and Roujean, J.-L.: Analysis of MODIS albedo changes over steady woody covers in France during the period of 2001–2013, *Remote Sens. Environ.*, 191, 13–29, doi:10.1016/j.rse.2016.12.019, 2017.

The variables chosen are key variables for the biophysical processes which determine the climate effect of surface property changes following storm damage.

(12) Please discuss your results in Discussion. Some of the discussions are found in Methods, so it is better to move the parts to the Discussion section. I am not sure whether the first paragraphs are necessary especially at the beginning of Discussion. For example, gusts are not directly analyzed in this research. Is 6.3 Salvage logging really important for your research, although there are no indications in the results? I think in the first parts of Discussion, using the same order of Results would help readers to easily understand your discussions.

Thank you for this comment. We agree that the section on salvage logging should be moved to the methods. Note that the simulations will benefit by the implementation of salvage logging because the presence/absence of salvage logging decouples/couples storm damage to insects outbreaks. Damaged woods due to storms are often left on site in unmanaged forests, however salvage logging is often applied in managed forest in order to recover some of the economic losses and to avoid large scale insects outbreaks triggered by wind disturbance. We will add this information to this paragraph and move the paragraph to the methods.

The referee is correct in stating that gusts are not directly analyzed in this study but we find it justify to bring up the difficulties in calculating gusts in the discussion. Estimating gustiness is a key challenge in wind damage studies and our solutions to this issues are far from final. We feel this information should not be limited to the method section. This is also a straightforward comment to address. Given the disagreement between the referees and our own position on this issues, we leave it to the editor to make the final suggestion.

---

## Author Comment (AC3) · 20 Nov 2017

Dear Executive Editor Lutz Gross,

I would like to thank you for your comment on uploading the model code. The link (https://forge.ipsl.jussieu.fr/orchidee/browser/branches/ORCHIDEE-DOFOCO/ORCHIDEE?rev=4262) directs to where the code used in this manuscript, which is archived.

Best regards,

Yi-Ying Chen

---

## Author Comment (AC4) · 12 Dec 2017

Dear Editor Lutz Gross,

The model code of ORCHIDEE-CAN in reversion 4262 including the windthrow module has been uploaded to a on-line repository provided by Zenodo with doi:10.5281/zenodo.1109750.

This repository provides another opportunity for readers or reviewers to browse the model code which can be accessed in the following web-link:

https://zenodo.org/record/1109750

The on-line code tracking/browsing service for the ORCHIDEE DOFOCO branch is still supported by IPSL via the following web-link:

https://forge.ipsl.jussieu.fr/orchidee/browser/branches/ORCHIDEE-DOFOCO/ORCHIDEE?rev=4262

with best regards,

Yi-Ying Chen on behalf of the authors

———————————————————

---

## Author Response (AR2)

**Responses to the Referee #1 : General Comment**

The authors incorporate a well-established wind disturbance model (ForestGALES) into a dynamic global vegetation model, the ORCHIDEE-CAN. It is perhaps the first study of windthrows simulation by an Earth System Model (ESM). I emphasize the novelty of this study because it improves our understanding of an overlooked agent of tree mortality (wind) in forest ecosystems.
Thank you for your kind remark.

**Responses to the Referee#1 : Specific concerns**

(1) Winds are a major agent of tree mortality, a well-known fact that has been discussed extensively in the literature over a range of spatial scales and ecosystems. Yet, the introduction of this study is very limited and does not justify why windthrows need their own representation scheme in an ESM. Furthermore, there is not a formal definition of wind storms. Wind storms can vary from strong winds to tropical cyclones. The frequency and the spatial scales of these events justify this study. However the reader is left to wonder whether this type of study is important.
We agree that wind storms are not properly defined in the manuscript and we added the definition of wind storms in the revised manuscript (**P3, L9-14**). We feel that the importance of simulating wind storm damage is addressed in the discussion and therefore moved this section forward and present it in the introduction (**P2, L10-22**)

(2) The use of a sigmoid function to represent the storm damage (Equation 9) was not justified.

Although different damage function with different dependencies could be conceived, for example, soil moisture dependent, topographic dependent and wind speed dependent, we considered that the wind speed variation is the most important factor in controlling the storm damage in a forest stand. We followed Anyomi et al. (2017) who used and validated a wind speed dependent approach for various tree species in the temperate region. We cited this paper in our revised manuscript as a justification for using this type of wind damage function to estimate the storm damage rate (**P8, L19-21**)

Reference: Anyomi, K. A., Mitchell, S. J., Perera, A. H., and Ruel, J. C.: Windthrow dynamics in Boreal Ontario: A simulation of the vulnerability of several stand types across a range of wind speeds, *Forests*, 8, 1–15, doi:10.3390/f8070233, 2017.

(3) The calculation of critical wind speed uses five tree species. How representative are these species of the whole simulation domain? Are these tree from the same Genus or Family?

Spruce, pine and birch make up almost the entire forest cover of Southern Sweden. Pine is the single most important species in Les Landes. These regions were only used to test the model. The simulation domain of this new development is Europe. The five species for which the model was tested make up 67% of the European forest cover. In terms of taxonomic families the representativeness increases to >90% (Novel Maps for Forest Tree Species in Europe, Renate Köble and Günther Seufert). We added this information in our revised manuscript (**P11, L9-10**)

Reference: Köble R. and Seufert G.: Novel maps for forest tree species in Europe. Proceedings of the 8th European Symposium on the Physico-Chemical Behaviour of Air Pollutants: "A Changing Atmosphere!", Torino (It) 17-20 September 2001.

(4) The comparison of modeled damage versus the observational data would benefit from the inclusion of percentage.

We agreed and added the relative model simulation errors ((estimation-observation)/estimation) to the figures for the comparison of modeled damage versus the observational data. The new figures (Figs 5 & 6 in the revised manuscript, **P31 & P32**) were shown as below.

[Figure]

Fig. 5. Sensitivity of the simulated storm damage over Sweden between 1981 and 2000 for different values for the relaxation parameter ($R_f$) and the gust factor adjustment $G_{adj}$ (A). Observed storm damage is extracted from (Nilsson et al., 2004; Schlyter et al., 2006; Bengtsson and Nilsson, 2007; Gardiner et al., 2010). The relative model simulation error ((estimation-observation)=estimation) for the best tuned case (B).

[Figure]

Fig. 6. Comparison of the storm damage simulated by the ORCHIDEE-CAN and the annual wood damage over the Sweden from 1951 to 2010. Observed storm damage is extracted from (Nilsson et al., 2004; Schlyter et al., 2006; Bengtsson and Nilsson, 2007; Gardiner et al., 2010) The dashed-line area is the period from 1981 to 2000, which was selected for parametrization. The RMSE of the estimated storm damage is $1.35 \times 10^6 m^3$ for the parametrization period and $5.05 \times 10^6 m^3$ during the evaluation period. The validation period ranges from 1951 to 2010 but excludes the years 1981 to 2000. The relative model simulation error for the validation period from 1981 to 2010 (B).

(5) Critical wind speed and downscaling require more detail. What is the wind speed needed to overturn trees in the study area? How does it compares with the critical wind speed?

We propose to increase the level of detail in the revised manuscript by showing the

modelled critical wind speeds for inner and outer forest areas and compared these CWS with daily maximum wind speed (**P15, L4-8; P15, L12-14**). We then used the contour lines to show the spatial distribution of the difference between minimum CWS and daily maximum wind speed (Fig. 4 in our revised manuscript, **P30**). The new figure is shown below.

[Figure]

Fig. 4 The ORCHIDEE-CAN calculated lowest critical wind speeds for overturning or stem breakage for forest located near to (inner) and away (outer) from a forest edge. When making the display, the critical wind speeds from the three diameter classes and four age groups from *Picea* species were assessed and the lowest value was compared against the daily maximum wind speed for estimating the damage due to storms. Lowest critical wind speeds in the forest away from a forest edge (outer) (A), lowest critical wind speeds in the forest near to a forest edge (inner) (B), lowest critical wind speeds overlaid with the difference between maximum daily wind speed and lowest critical wind speed in outer area on 9th January, 2005 (C), lowest critical wind speeds overlaid with the difference between maximum daily wind speed and lowest critical wind speed in the outer area (D). The contours show the positive wind speed difference in black and the negative wind speed difference in red. Forests within the red contours are expected to suffer from storm damage.

(6) An analysis of forest damage focusing on pixel heterogeneity vs wind speed is relevant for this study.

It is unclear to us what the referee had in mind when making this comment. Nevertheless, we realize that the current implementation overlooks many sources of sub-pixel heterogeneity when calculating storm damage but we would like to stress that one of the novelties of this study is that we found a solution (i.e., the calculation of an inner and outer area) to use the stand-level ForestGALES at a much larger scale while avoiding the need to increase the spatial resolution (and thus the computational costs) of ORCHIDEE-CAN. Following a comment by the other referee, this novelty has be stressed in the methods, discussion and conclusion.

Further improvements towards accounting for sub-pixel heterogeneity are the topic of ongoing funding applications.
The article entitles, "Simulating damage for wind storms in the land surface model ORCHIDESS-CAN" is aimed to develop an earth system model using the submodel, ORCHIDESS-CAN, with focusing on storm wind damage in forests. In terms of wind damage, a mechanistic-empirical model, ForestGALES, was coupled with ORCHIDESS-CAN. I understand that developing such models is really important to understand the environmental system and to make strategies for climate change. However, I think the scientific originality in this article seems weak partly due to the writing style. At this moment, this article is not well organized. The structure of the method section need to be improved. The result section is hard to understand and some paragraphs should be located in Discussion. The results are not well discussed in Discussion. In addition, comparison between calculated and observed data requires some statistical analysis in order to show how much reliable what you did. Some ambiguous expressions were found in Results.

In our revised manuscript, we paid attention to the structure of the manuscript. Given the aim of GMD, we decided to focus this manuscript on presenting how ORCHIDEE was further developed to accommodate the principles of ForestGALES. The new version of ORCHIDEE was tested (reported in the results) but given the scope of the journal the effects of wind damage over the test regions were not discussed in any detail. If we would have had access to other data, we could have picked any other region in the world to test the model. In other words, we have no special interest in storm damage

in Sweden or France. It just happened that due to our professional network we had access to good validation data for these regions. Therefore, it was decided to focus the discussion on the model philosophy and its possible improvements. In the future we would like to use ORCHIDEE-CAN+ForestGALES to simulate the storm damage at a large scale, not just Southern Sweden and Southwestern France.

When using existed models in research, it is very important to show what the new ideas and findings are.

We agreed with the referee that this aspect is insufficiently developed in the manuscript. In our revised version of the manuscript we stressed the following novelties:

- Almost the same group of authors developed the model capability to simulate forest and canopy structure. Although these developments are already published they are essential for the work presented here.
- In this manuscript we suggest a numerically efficient solution to deal with one of the key aspects of sub-pixel heterogeneity, i.e., recent forest edges.
- In this manuscript we went beyond the capabilities of ForestGALES by adding and parameterizing a damage function.
- In this manuscript we suggest a solution to account for aggregated driver data which are typically used in large scale simulations.

These novelties have been listed in the abstract (**P1, L6-10**) and the conclusions (from **P19, L22 to P20, L3**). In the methods section we indicated whether the presented method comes from GALES or was developed as part of this study (see section 2.1 to section 2.2 in the revised manuscript).

**Responses to the Referee #2:Specific concerns**

(1) The section of Methods should be improved. I think the structure and components need to be modified. For example, first models used in the article are explained including the parameters and equations. Second, what your original ideas are explained. A figure (diagram or flowchart) would be helpful to show the process. Third, validation procedures can be explained including the input data of three regions with references. Also please explain why you chose the regions and different analysis were conducted for them.

If we correctly understand the reviewer, the suggestion is to merge the information currently presented in section 2 (models) and 3 (parameters). In our revised manuscript, the model description and parametrization were merged into a single section (section

2). As mentioned above we indicated which developments are new and which were taken from ForestGALES. A flowchart linking all elements presented in the methods could look like the figure shown below (**P5, L9-14**).

[Figure]

Fig. 1. Information flow of this study showing the link between the different elements presented in sections 2.2 and 2.3 (numbers in the figure refer to section numbers in the text). The diagram shows input data in blue color. The dashed box shows critical wind speeds calculations according to ForestGALES.

We are confused by the suggestion to make a third section that presents the validation procedures and the available input data for the test cases as we believe that is exactly what is done in section 4. We revised the manuscript to better structure the information previously contained in sections 2, 3 and 4.

(2)  Please explain what ORCHIDESS-CAN can do and how to use the model with required inputs. Is ORCHIDESS-CAN a grid-based model? Can we change the grid size (resolution)? Has the structural growth model been included in the model? How did you exactly integrate ORCHIDESS-CAN and ForestGALES? Did you make new codes? I think a diagram would be helpful to understand how the models work and what you actually did in the study.

In section 2 we added a description of the basic principles of ORCHIDEE (**P3, L20 to P4, L13).** We made new code to consider the physic of ForestGALES to ORCHIDEE-CAN and the new model implementation/code can be downloaded via web-link: https://doi.org/10.5281/zenodo.1109750 (**P20, L15-16**).

(3) I suppose that you used some observed data such as satellite photos and forest damage data. But some of the data sources were not indicated in Methods.

Thanks for pointing out this issue. When preparing the manuscript we indeed overlooked adding a detail description of the observational data. We added more information about the in-situ observations. This information has been added into an extra section 3.1 (**P11, L13-24**).

(4) In terms of the equations, I do not understand why 9h needs to be doubled in eq. How did you calculate the average mean tree height and distance between trees? Are they averaged in a single grid? What data was used to calculate them?

Figure S1 shows the concept of inner and outer area generated by a gap created by human disturbance in a managed forest. Given the suggestions by both reviewers, the revised manuscript will already contain 11 items (9 figures + 2 tables), therefore, we prefer not to add this figure in the revised manuscript as it can be available through the supplementary (**Figure S1**).

[Figure]

Fig. S1. Illustration of inner and outer area generated by gaps within a forest area.

We calculated the dynamics of forest gap area in a modeled grid by making use of historical land use/cover maps compiled by McGarth et al. (2015). This part of description can be found in the revised manuscript (**P12, L6-8**).

Reference: McGrath, M. J., Luyssaert, S., Meyfroidt, P., Kaplan, J. O., Bürgi, M., Chen, Y., Erb, K., Gimmi, U., McInerney, D., Naudts, K., Otto, J., Pasztor, F., Ryder, J., Schelhaas, M.-J., and Valade, A.: Reconstructing European forest management from 1600 to 2010, Biogeosciences, 12, 4291-4316, https://doi.org/10.5194/bg-12-4291-2015, 2015.

(5) In this article a term, actual wind speed, is quite often found. However, I do not think we can obtain actual wind speed data except maybe around an anemometer. The wind speeds used here would be "estimated" wind speeds.

Fair point. Given that also the critical wind speeds are estimated we propose to simply use "wind speed" and "critical wind speed" throughout the manuscript.

(6) In results, some methods and terms such as spatio-temporal comparison, Beaufort wind scale, and root-mean-square error are not explained in Methods.

The term of "spatio-temporal" was replaced by "spatial and temporal aggregation" (**P14, L3; P17, L18**). The reference for the Beaufort wind scale classification (**P10, L23-24**) was added and the statistical index for evaluating the model error, was also added into the revised manuscript (**P10, L26-30**).

(7) Please try to write the results as simple as possible. Some sentences and paragraphs seem to be unnecessary or better to move to Discussion. For instance, I do not think the first sentence on page 12 is important in Results. The second and fourth paragraphs on page 13 should be located in Discussion.

Although we agree with the referee that part of the material presented in section 5.4 could also be presented in the discussion, we though and still think that it makes a stronger manuscript if the test cases are not promoted to the topic of the discussion. Passing the tests is just a minimum requirement. The test themselves are not the topic of the manuscript. The topic of the manuscript is the model and therefore our discussion focusses on the model.

We consider the first sentence on page 12 to be important because it justifies why we believe that a model-model comparison is, in this case, a valid surrogate for a more

traditional model-data comparison.

The changes proposed by the referee are straightforward to implement. Given that the other referee was not disturbed by which material was presented in the results and which material was presented in the discussion, we leave it to the editor to make the final suggestion.

(8) In Figure 1, the mean wind ratios of BWSs 8, 9, 10, and 11 should be explained. Although a fitting line was created, I am not sure how it is reliable. For example, if BWS 11 or BMS 9 are outliners, the line will become different. Why are wind speeds in BWS 10 lower than those in BWSs 8 and 9? To use the fitting line for the following analysis, it is important to justify how much statistically suitable the line is.

We fully agree with this suggestion. The Figure1 (in the original manuscript) shows the effect of the spatial and temporal aggregation from CRU-NCEP 6h-halfdegree wind speed to Fluxnet 30 min plot scale wind speed. We applied a 4th-order nonlinear equation to capture the variation of MWR by fitting the mean value of MWR in each BWS scale from scale 1 to scale 11. Although, we didn't show the effect of temporal aggregation by analyzing the Fluxnet wind fields, we did explore the temporal aggregation effect on the Fluxnet wind fields by making use the approach suggested by Larsen and Mann (2006). By using different averaging periods (m=2 for 1h, m=4 for 2h, m=8 for 4h and m=12 for 6h, etc.) the variation of MWR gradually saturated to a constant value which is not very sensitive to the low wind speed ranging from BWS3 to BWS6, but this value increases dramatically for high wind speed (i.e. from BWS9 to BWS11) (see the figure below). We added this information to the SI as Figure S3 to support this approach. This analysis suggests that BWS10 is the outlier, possibly due to the fact that all data come from a single site. We added this reasoning to **P14, L6-8** in our revised manuscript.

[Figure]

Fig. S3. The effect of different temporal aggregation period, an increasing in *m* index for a 30 min extension, on the wind speed at selected Fluxnet forest sites. The red line in each sub-plot indicates of using a 6h aggregation period (m = 12, because m = 1 represents 30 minutes).

When adding the effect of the temporal aggregation derived from Fig. S3 to Fig 2 in the revised manuscript the figure would look as follows.

[Figure]

Fig. 2. Distribution of the mean wind ratio (MWR) in each Beaufort wind scale (BWS) and the relationship between the six hour CRU-NCEP reanalysis wind speed and MWR. Fitting of the relationship (red line) used Eq. with regression coefficients $a_0$=-5.299, $a_1$=2.051, $a_2$=-0.191 and $a_3$=0.006. This relationship is used to convert CRU-NCEP six hour mean wind speed to the 30min maximum wind speed in this study. The RMSE of using this regression model to predict the mean value of MWR in each BWS class is 0.48. The open circles (gay color) show the effect of 6h temporal aggregation on the MWR from the selected Fluxnet sites in European forest. The gray line is the fitting line of the open circles.

The effect of temporal aggregation on MWR shows a similar nonlinear pattern to the equation that we applied for downscaling the wind fields. We also reported the RMSE of the nonlinear regression line to the mean MWR values to justify the model error.

Reference: Larsen, X. G. and Mann, J.: The effect of disjunct sampling and averaging

time on maximum mean wind speeds, *J. Wind Eng. Indust. Aerodyn.*, Vol.94, pp.581-602, 2006.

(9) In terms of Figure 2, are there comparisons between the critical wind speeds calculated by ForestGALES only and those by ORCHIDESS-CAN + ForestGAELS? If so, are the differences only from input data to the models? In addition, to compare the speeds between stem breakage and uprooting, it is important to conduct some statistical analysis in order to show how statistically different or similar between the two outputs are. Which graphs are for the forest edges?

Figure 2 (Figure 3 in the revised manuscript) summaries the results of CWSs calculations both form ORCHIDEE-CAN and ForestGALES with the same canopy structure variables, which were simulated by ORCHIDEE-CAN. (Reviewer can refer to the context in the section 3.3 in the revised manuscript.) In order to evaluate the difference of CWSs calculation between two models, we applied Eq. (11) to calculate RMSE, in which CWSs calculated from ForestGALES were treated as references (**P14, L20-25**). CWSs for the forest edge are shown in upper panels (from (A) to (E)) having lower CWSs comparing to CWSs for the area away from the forest edge in lower panels (from (F) to (J)).

[Figure]

Fig. 3. Model calculated critical wind speeds as a function of tree height for five common tree species in Europe. For each tree species the critical wind speed is calculated for overturning and stem breakage for forest located away from a forest edge (outer, upper panels A-E) and nearby a forest edge (inner, lower panels F-J). The

ORCHIDEE-CAN simulations are shown as symbols and benchmarked against the ForestGALES simulations which are shown as lines. The scientific names of the tree species are given in section 2.3.2. The CWSs difference between the ORCHIDEE-CAN and ForestGALSE calculation were calculated using Eq. (11), which the CWSs from ForestGALSE were treated as the reference.

(10) In Figure 3, I think the lowest critical wind speeds are more important to consider wind damage rather than the averaged critical wind speeds. Also, are these results for the current forest conditions?

We replaced the current Figure 3 by a new Figure 4 (**P30** in the revised manuscript, see above) which only shows the lowest critical wind speed for inner and outer area. This information is then overlaid with the difference between the lowest critical wind speed and the maximum daily wind speed. These results are for the forest condition on 9th January, 2005. Note that the model only uses the lowest critical wind speed (and not the average) to calculate storm damage. As noted by the referee, the figure was inconsistent with the model code and model approach. This inconsistency has now been corrected.

(11) In terms of the results from Les Landes, I do not understand why these indicators were chosen to represent the results although they seem to be interesting. It is important to clearly address in Methods what kind of analysis would be conducted in advance. Why 20 values were used in (A) and (B) in Figure 8? Where these values came from?

We complied the remote sensing data used from the study by Planque et al. (2017) the selected points are shown in pink arrows and the ORCHIDEE simulation grids are overlaid in black lines. We then ran the model for ten years from 2001 to 2010 and extracted the ORCHIDEE outputs from two selected pixels in white box, one near the eastern part of Les Landes forest and another near the middle of Les Landes forest.

[Figure]

Fig. S2. Illustration of selected points in the les lands forest.

Although, the temporal and spatial resolution from remote sensing and model simulation are different, we merged these two datasets into ten year summer-time slots in two locations. This is the reason for having 20 points in the scatter plot of comparing the remote sensing observation and model simulation. We added this part of description into the section 3.1 **(P11, L23-24)** and added this in the supplementary as Figure S2**.**

Reference: Planque, C., Carrer, D., and Roujean, J.-L.: Analysis of MODIS albedo changes over steady woody covers in France during the period of 2001–2013, *Remote Sens. Environ.*, 191, 13–29, doi:10.1016/j.rse.2016.12.019, 2017.

The variables chosen are key variables for the biophysical processes which determine the climate effect of surface property changes following storm damage.

(12) Please discuss your results in Discussion. Some of the discussions are found in Methods, so it is better to move the parts to the Discussion section. I am not sure whether the first paragraphs are necessary especially at the beginning of Discussion. For example, gusts are not directly analyzed in this research. Is 6.3 Salvage logging really important for your research, although there are no indications in the results? I think in the first parts of Discussion, using the same order of Results would help readers

to easily understand your discussions.

Thank you for this comment. We agree that the section on salvage logging should be moved to the methods. Note that the simulations will benefit by the implementation of salvage logging because the presence/absence of salvage logging decouples/couples storm damage to insects outbreaks. Damaged woods due to storms are often left on site in unmanaged forests, however salvage logging is often applied in managed forest in order to recover some of the economic losses and to avoid large scale insects outbreaks triggered by wind disturbance. We added this information to this paragraph and move the paragraph to the methods (section 2.2.5; **P9, L2-14**). We restructured our manuscript and added a new discussion section, section 5.2 (from **P17, L31** to **P18, L17**), for the discussion of storm damage over Sweden.

The referee is correct in stating that gusts are not directly analyzed in this study but we find it justify to bring up the difficulties in calculating gusts in the discussion. Estimating gustiness is a key challenge in wind damage studies and our solutions to this issues are far from final. We feel this information should not be limited to the method section. This is also a straightforward comment to address. Given the disagreement between the referees and our own position on this issues, we leave it to the editor to make the final suggestion.

[revised manuscript text omitted]